# PITSTOP: PHYSICS-INFORMED TRAINING WITH GRADIENT STOPPING

## ABSTRACT

Physics-informed learning offers a powerful approach for modeling physical systems by enforcing governing equations directly within the training process. However, optimizing such models remains inherently challenging, especially for large systems, due to the ill-conditioned nature of these residual-based loss functions. In this paper, we critically examine the limitations of classical optimization techniques by developing a comprehensive theoretical framework for physics-informed setups, including insights on convergence guarantees, convergence speed and fixed points. Next, we introduce PitStop, a novel optimization method for physics-informed training based on gradient stopping, which overcomes the limitations of classical methods by backpropagating feedback differently from the standard chain rule of calculus. The method is motivated and mathematically analyzed in our theoretical framework, incurs no additional computational cost compared to standard gradients, and achieves superior results in our experiments. Our work paves the way for more scalable and reliable physics-informed model training by fundamentally rethinking optimization paradigms.

## 1 INTRODUCTION

Physics-informed objectives enable the learning of physical models in a fundamentally different way from supervised learning. By minimizing the residuals of the governing equations of the system, the model learns solutions that are consistent with the underlying physics, even in regions where data is sparse, noisy, or unavailable, as demonstrated in various works Raissi et al. (2017); Mao et al. (2020); Karniadakis et al. (2021).

Since their introduction, these training setups have been notoriously challenging to optimize. As the system size increases, the residual form of the loss function becomes highly ill-conditioned Krishnapriyan et al. (2021); Wang et al. (2021), making gradient-based methods slow due to poor update directions. While second-order optimizers are often employed to address this issue Nocedal & Wright (1999), their updates are computationally expensive, resulting in an overall approach that remains slow and leaves the problem inadequately resolved.

In this work, we challenge this classical optimization perspective as a fundamentally flawed way of approaching physics-informed setups. The core issue lies in the fact that, while a perfect zero-loss minimum of the physics-informed loss is also a perfect minimum of the supervised loss, an approximate minimum of the physics-informed loss does not necessarily correspond to an approximate minimum of the supervised loss. Therefore, the steepest descent property of gradient descent becomes ineffective in the typical machine learning setting where solutions are approximate.

To address this problem, we propose a method that backpropagates feedback from the physics-informed loss in a manner that deviates from the standard gradient computation by omitting certain terms of the chain rule, a technique often referred to as gradient stopping. While such operations have previously been used to shorten computation graphs and reduce gradient magnitudes to aid optimization, our approach introduces a fundamentally different mechanism that goes beyond the classical optimization paradigm.

We present the following contributions in this work:

- *PitStop*, a physics-informed training technique based on gradient stopping that transcends the classical scope of optimization methods.
- A comprehensive theoretical framework that analyzes PitStop through the lens of linear iteration schemes, focusing on convergence criteria, convergence speed, and fixed points.
- Experiments that validate our theoretical results both within and beyond its formal bounds.

## 2 PROBLEM SETUP

**Differential equations** We investigate the optimization of a function approximation model $y$, whose input-output mapping directly corresponds to the solution of a differential equation. Using $t$ and $x$ for temporal and spatial variables, respectively, $y$ acts from $[0, \mathcal{T}] \times \mathcal{D} \to \mathbb{R}^{n_y}$ with $(t, x) \mapsto y(t, x)$. Here, $\mathcal{T}$ is the end of the time interval, $\mathcal{D} \subset \mathbb{R}^{n_x}$ is the spatial domain, and lastly, $n_x$ and $n_y$ denote the dimension of spatial and field variables $x$ and $y$, respectively. To keep the notation compact, we omit indicating the dependence of $y$ on the parameters $\theta$ of the function approximation model. We consider time-dependent differential equations with a spatial differential operator $\mathcal{P}$, which can be a function $\mathcal{F}$ of $y$ and all its spatial derivatives.

$$\partial_t y = \mathcal{P}(y) = \mathcal{F}(y, \partial_x y, \partial_x^2 y, ...) \tag{1}$$

As an example, if $\mathcal{F}$ contains only $y$ itself and not its spatial derivatives, this setup reduces from a partial differential equation to an ordinary differential equation. Additionally, we incorporate boundary conditions by introducing $y^{\text{init}}$ and $y^{\partial \mathcal{D}}$ as follows:

$$
\begin{aligned}
y(0, x) &= y^{\text{init}}(x) && \forall x \in \mathcal{D} && \text{(Initial condition)} \\
y(t, x) &= y^{\partial \mathcal{D}}(t, x) && \forall (t, x) \in [0, \mathcal{T}] \times \partial \mathcal{D} && \text{(Spatial boundary condition)}
\end{aligned}
\tag{2}
$$

**Loss functions** Physics-informed training means we train directly against the residual $\partial_t y - \mathcal{P}$ of the differential equation, as opposed to supervised methods, which require an explicit solution $y^{\text{sol}}$ for training. Nevertheless, the supervised loss is important for evaluation, as we are ultimately interested in how close we are to the actual solution. Physics-informed (PI) loss and supervised (SV) loss are given as follows, with $\|\cdot\|$ being the L2 norm:

$$\mathcal{L}_{PI} = \underbrace{\int_{[0,T] \times \mathcal{D}} \left\| \partial_t y - P(y) \right\|^2}_{\text{Interior residual term}} + \alpha \underbrace{\int_{\mathcal{D}} \left\| y|_{t=0} - y^{\text{init}} \right\|^2}_{\text{Initial condition}} + \beta \underbrace{\int_{[0,T]} \left\| y|_{\partial \mathcal{D}} - y^{\partial \mathcal{D}} \right\|^2}_{\text{Spatial boundary}} \tag{3}$$

$$\mathcal{L}_{SV} = \underbrace{\int_{[0,T] \times \mathcal{D}} \left\| y - y^{sol} \right\|^2}_{\text{Interior supervised term}} + \underbrace{\int_{\mathcal{D}} \left\| y|_{t=0} - y^{\text{init}} \right\|^2}_{\text{Initial condition}} + \underbrace{\int_{[0,T]} \left\| y|_{\partial \mathcal{D}} - y^{\partial \mathcal{D}} \right\|^2}_{\text{Spatial boundary}} \tag{4}$$

In $\mathcal{L}_{PI}$, we introduce two constants, $\alpha$ and $\beta$, that can be used to adjust the weighting of the boundary terms as required. Our study presents a method to optimize $\mathcal{L}_{SV}$ by constructing parameter updates derived from the computation of $\mathcal{L}_{PI}$, which is the quintessential physics-informed training task.

**Setup** We restrict ourselves to a setup where the time derivative is discretized directly through a numerical time-stepping scheme, rather than being computed via automatic differentiation. This is because the method we propose requires temporal discretization. Secondly, we use fixed collocation points during training to work within a deterministic setup. This allows for a closer comparison between theory and experiments by eliminating sources of stochasticity, and therefore allows for more conclusive results. Together, these two conditions provide a reduced framework for study, which we use to reproduce the optimization difficulties of physics-informed tasks and demonstrate our solution. This approach emphasizes that the problems stem from the different form of the loss—residual-based rather than supervised—and not from how the loss is sampled or whether derivatives are computed via automatic differentiation.

Implementing these two conditions works as follows: We split the temporal and spatial domains into sequences of $n_i$ and $n_j$ equidistant points, respectively, denoted $\{t_i\}_{i=0}^{n_i}$ and $\{x_j\}_{j=0}^{n_j}$. Next,

**Algorithm 1** PitStop

---

1: Initialize collocation points $p$, model $M$ with parameters $\theta$, initial condition $y^{\text{init}}$ with weighing factor $\alpha$, explicit $P_E$ and implicit $P_I$ parts of the time evolution scheme, and learning rate $\eta$
2: **repeat**
3:      $L \leftarrow 0$                                               ▷ reset physics-informed loss
4:      $y \leftarrow M(p)$                               ▷ compute field values from model
5:      $L \leftarrow L + \alpha \cdot \|y[0] - y^{\text{init}}\|^2$         ▷ add initial condition error
6:      **for** $i = 1 \dots t$ **do**
7:          $E \leftarrow y[i-1] + P_E(y[i-1])$              ▷ explicit part
8:          $I \leftarrow y[i] - P_I(y[i])$                 ▷ implicit part
9:          $\tilde{E} \leftarrow \text{StopGradient}(E)$       ▷ stop backpropagation through $E$
10:         $L \leftarrow L + \|I - \tilde{E}\|^2$                ▷ add residual to loss
11:      **end for**
12:      $u \leftarrow \text{Backpropagate}(L, \theta)$        ▷ backpropagate feedback
13:      $\theta \leftarrow \theta + \eta \cdot u$                   ▷ update model parameters
14: **until** $\theta$ converged

---

we collect all evaluations of $y$ at time $t_i$ into a vector $y_i$ defined by $(y_i)_j = y(t_i, x_j)$, and do the same for $(y_0)_j = y^{\text{init}}(x_j)$ and $(y_i^{\text{sol}})_j = y^{\text{sol}}(t_i, x_j)$. For the spatial differential $\mathcal{P}$, which acts on continuous functions, we introduce the corresponding operator $P$ for the discrete setting. Additionally, we collect the spatial boundary terms also into $P$ to simplify the notation for our core ideas, which primarily concern the temporal direction. Along this dimension, discretizing the time derivative results in $n_i$ residual terms $R_i$, each involving terms at two consecutive time points $y_i$ and $y_{i-1}$:

$$R_i = \left(1 - P_I\right)(y_i) - \left(1 + P_E\right)(y_{i-1}) \tag{5}$$

Here, $P_I$ and $P_E$ stand for the implicit and explicit parts of the time evolution scheme. For instance, for a time step of duration $\tau$, choosing $(P_I, P_E)$ as $(0, \tau P)$ results in the explicit Euler scheme, while $(\tau P, 0)$ gives the implicit Euler scheme, $(\tau P/2, \tau P/2)$ for the semi-implicit Euler scheme, or involving powers of $P$ results in multi-stage Runge-Kutta methods. Finally, denoting the discretized initial condition by $R_0$, we can compactly express our loss functions of interest in the discrete setting as:

$$L_{PI} = \alpha \|R_0\|^2 + \sum_{i=1}^{n_i} \|R_i\|^2 \qquad\qquad L_{SV} = \sum_{i=0}^{n_i} \left\|y_i - y_i^{\text{sol}}\right\|^2 \tag{6}$$

## 3 METHOD

Based on this setup, we investigate iterative methods where an update direction $u$ is computed from the current parameters $\theta$ and is multiplied by a learning rate $\eta$ to update the parameters, leading to $\Delta\theta = -\eta u(\theta)$. Our method PitStop (PS) defines the update direction as:

$$u_{PS} = \alpha \cdot (\partial_\theta y_0)^\dagger \cdot R_0 + \sum_{i=1}^{n_i} (1 - \partial_y P_I)(\partial_\theta y_i)^\dagger \cdot R_i \tag{7}$$

To understand this update form, let us compare PitStop to the Gradient Descent (GD) direction:

$$u_{GD} = \alpha \cdot (\partial_\theta y_0)^\dagger \cdot R_0 + \sum_{i=1}^{n_i} \left( \left(1 - \partial_y P_I\right)(\partial_\theta y_i) - \left(1 + \partial_y P_E\right)(\partial_\theta y_{i-1}) \right)^\dagger \cdot R_i \tag{8}$$

The initial condition is processed in the same way, but the residual term of PitStop lacks the chain rule terms of the earlier predictions. The key motivation is that, in an initial value problem, errors in the residual term ought to affect only subsequent predictions, not past ones. Such an update procedure can be implemented using standard gradient stopping operations, which allow to omit certain backpropagation paths during a gradient call, making PitStop updates straightforward to implement and even cheaper to compute than full gradients. The implementation procedure can be

found in Algorithm 1. A second relevant update direction is Gauss-Newton (GN):

$$
u_{GN} = \begin{pmatrix} \partial_\theta y_0 \\ \left(1 - \partial_y P_I\right)(\partial_\theta y_1) - \left(1 + \partial_y P_E\right)(\partial_\theta y_0) \\ \vdots \\ \left(1 - \partial_y P_I\right)(\partial_\theta y_{n_i}) - \left(1 + \partial_y P_E\right)(\partial_\theta y_{n_i-1}) \end{pmatrix}^{-1} \cdot \begin{pmatrix} \alpha R_0 \\ R_1 \\ \vdots \\ R_{n_i} \end{pmatrix}
\tag{9}
$$

Its distinctive feature is the inversion operation, which is computationally expensive but yields a high-quality update direction. Between Gradient Descent and Gauss-Newton, all the classical derivative-based optimization methods are located, each offering a different trade-off between a good update direction and computational cost.

On an intuitive level, PitStop can be understood from a causal perspective. In initial value problems, earlier states influence later ones, not the reverse. Consequently, when a physics-informed residual $R_i$ is nonzero and the later state $y_i$ in that residual term fails to match the dynamics implied by the earlier state $y_{i-1}$ in that residual, then the optimization algorithm should modify only the later state. This is what PitStop does through gradient stopping. We will now classify the mathematical features of PitStop, showing that it is a profoundly different method going beyond this classical spectrum of optimization methods.

## 4 THEORY

Our theoretical analysis builds on linear fixed-point theory, a standard framework for iterative methods. We extend it by i) incorporating a second loss function that does not affect updates, and ii) generalizing to non-symmetric fixed-point operators. This allows application to physics-informed objectives, where the supervised loss acts as a second loss, and to analyzing our method, PitStop.

We start from Banach's fixed point theorem, which guarantees convergence of $\theta_{k+1} = H(\theta_k)$ to a unique fixed point $\theta^*$ if $H$ is contracting. In the linear case, $H$ is a matrix and contraction means its operator norm is less than 1. Linear insights extend to the nonlinear case because near any fixed point, the function is locally linear. Thus, mechanisms causing divergence linearly also hinder nonlinear convergence, highlighting linear analysis as a foundational step.

### 4.1 LINEARIZATION OF PHYSICS-INFORMED TRAINING

The update equations of GD, GN, and PS are linear if two conditions hold: i) the function approximator $y$ is linear in the parameters $\theta$, though the input-output mapping can still be nonlinear, $y_\theta(t,x) = \sum_{j=1}^{n_p} f_j(t,x)\,\theta_j$, where $n_p$ is the number of parameters; and ii) the differential equation and corresponding time evolution scheme are linear, so the residual can be written as $R_i = My_i - Ny_{i-1} - c_i$, with $M = 1 - \partial_y P_I$, $N = 1 + \partial_y P_E$, and $c$ collecting all remaining zeroth-order terms. For purely explicit or implicit schemes, either $M$ or $N$ reduces to the identity.

Under these assumptions, both physics-informed and supervised losses are quadratic in $\theta$, and the physics-informed loss becomes $L_{PI} = \|F\theta - c\|^2$. The update equations then simplify to the following, where $B$ encodes how feedback is backpropagated:

$$
\theta_{i+1} = \theta_i - \eta B(F\theta_i - c) = (1 - \eta BF) \cdot \theta_i + \eta Bc
\tag{10}
$$

In particular, $B = F^\dagger$ for GD and $B = F^{-1}$ for GN. For PS, an expression for $B$ is summarized in the following theorem, which will serve as the basis for our further analysis.

**Theorem 1: Update equation for PitStop**

*In the linear case, PitStop performs updates of the form $u_{PS} = B(F\theta - c)$, where:*

$$
\begin{aligned}
F = TA \quad &with \quad T = \sqrt{\alpha} \cdot E \otimes 1 + (1 - E) \otimes M - O \otimes N \\
B = A^\dagger U^\dagger \quad &with \quad U = \sqrt{\alpha} \cdot E \otimes 1 + (1 - E) \otimes M
\end{aligned}
\tag{11}
$$

*Here, $A$ denotes the row-stacked constant Jacobians $A_i = \frac{\partial y}{\partial \theta}\big|_{t_i}$, i.e. $A^\dagger = [A_0^\dagger \quad A_1^\dagger \quad \ldots \quad A_{n_i}^\dagger]$. Furthermore, $E \in \mathbb{R}^{n_j \otimes n_j}$ with $E_{ab} = 1$ if $a = b = 0$ and $E_{ab} = 0$ elsewhere, and $O \in \mathbb{R}^{n_j \times n_j}$ with $O_{ab} = 1$ if $a - b = 1$ and $O_{ab} = 0$ elsewhere, and $1$ are identity matrices of appropriate size.*

## 4.2 Convergence criterion

The iteration scheme in 10 converges if the fixed-point operator is contracting, i.e., $|1 - \eta BF| < 1$. For classical optimization methods, $BF$ is symmetric, so the condition reduces to $BF$ being strictly positive definite. If $BF$ had both positive and negative eigenvalues, the operator could not be contracting for any $\eta$. We assume $A$ to be full rank throughout, which can always be ensured by restricting to the subspace orthogonal to its null space. In particular, $BF = F^\dagger F$ for Gradient Descent, which is always strictly positive definite, ensuring convergence. Similarly, $BF = F^{-1}F = I$ for Gauss-Newton, which is also positive definite, so Gauss-Newton converges. Convergence analysis for PitStop is more involved; the main results are summarized in the following theorem:

**Theorem 2: Feature-independent convergence criterion for PitStop**

*Let $M$ be non-singular, and let $M$ and $N$ satisfy $\sigma_{\min}(M^\dagger M) > \sigma_{\max}(M^\dagger N)$, i.e., the smallest singular value of $M^\dagger M$ is strictly greater than the largest singular value of $M^\dagger N$. If the weight factor $\alpha$ for the initial condition term is chosen such that $\alpha > \sigma_{\max}(M^\dagger N)$, then PitStop converges.*

Recall from Theorem 1 that $BF = A^\dagger U^\dagger T A$ for PitStop. Intuitively, the conditions in Theorem 2 ensure that $U^\dagger T$ is positive definite in a geometric sense, which in turn guarantees that the eigenvalues of $A^\dagger U^\dagger T A$ have positive real parts and PitStop converges. These complication in the notion of positive-definiteness are due to the non-symmetry of the involved matrices and we discuss them in the proof of Theorem 2. Feature-independent convergence is crucial for practical use, allowing flexibility in architecture choice to improve generalization and reducing divergence mechanisms in nonlinear function approximation, where features vary across parameter space. Even if $P$ is not positive definite, convergence may still occur, but it becomes feature-dependent. Notably, the theorem's criteria depend only on properties of the time-stepping scheme ($M$ and $N$) yet guarantee convergence over arbitrarily long time horizons, connecting numerical analysis and learning principles. Finally, the theorem also includes a condition on the weighting of the initial condition, which classical methods often choose arbitrarily.

## 4.3 Convergence rate

The convergence rate $\rho$ measures how fast an iteration scheme approaches its fixed point, defined as the worst-case per-step decrease in distance to the fixed point along the optimization trajectory:

$$\rho = \lim_{k \to \infty} \max_{\theta_0 \in \mathbb{R}^{n_p}} \sqrt[k]{\frac{\|\theta_k - \theta^*\|}{\|\theta_0 - \theta^*\|}}. \tag{12}$$

This quantity depends on the learning rate. For Gradient Descent with optimal learning rate, $\rho$ depends on the condition number, with the number of iterations to reduce the distance by a factor scaling roughly linearly with it. Thus, Gradient Descent becomes increasingly ineffective for ill-conditioned tasks. In contrast, Gauss-Newton can optimize a quadratic loss in one step, but at higher per-iteration cost due to matrix inversion. Gradient Descent and PitStop, however, rely only on forward operations, making them feasible for large-scale optimization. Regarding PitStop's convergence rate, we find:

**Theorem 3: Improved condition number of PitStop**

*Assume PitStop converges. Define the condition number $\kappa^X(A) = \lambda_{\max}(A^\dagger X A)/\lambda_{\min}(A^\dagger X A)$ for $X$ full rank and $A$ the feature matrix. Assume a distribution $p(A)$ over relevant feature matrices $A$ and a model for condition number fulfilling $p(\kappa^X|A) = p(\kappa^X)$ when $X$ is symmetric. Then, PitStop has a lower condition number than Gradient Descent: $\mathbb{E}[\kappa^{U^\dagger T}] \leq \mathbb{E}[\kappa^{T^\dagger T}]$*

The condition-number model is motivated by the observation that, for large random feature matrices, the condition number depends more on matrix size than on matrix entries. Under this assumption we can relate the condition numbers of Gradient Descent's fixed-point operator and the symmetric part of PitStop's fixed-point operator, even though they differ. The proof itself relies on a theorem about the eigenspectra of positive definite, non-symmetric matrices, which states that adding a skew-symmetric component to the fixed-point operator clusters the eigenvalues more tightly, significantly improving the convergence rate of the associated iteration scheme. While this may seem counterintuitive, since a matrix's eigenspectrum is generally a complex function of its components, we note that the theorem does not assume specific eigenvalues, only that they are positive. It makes no

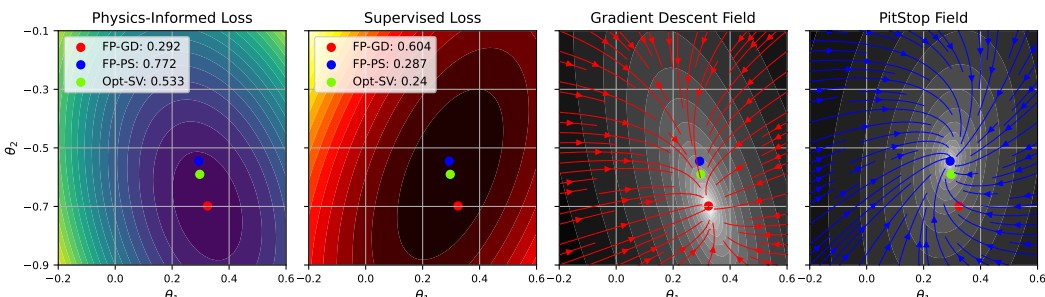

Figure 1: Toy example: fixed points of Gradient Descent (FP-GD), PitStop (FP-PS), and the supervised optimum (Opt-SV) with corresponding loss values in the legends: (a) physics-informed loss, (b) supervised loss, (c) gradient and (d) PitStop update vector field with background color indicating the update vector length

claims about the exact spectrum with a skew-symmetric part, providing instead a relative statement: the eigenspectrum becomes more balanced compared to its original distribution. This universal property explains why PitStop converges faster, even without knowing the exact spectrum of the fixed-point operator.

## 4.4 FIXED POINTS

An interesting aspect of our method is that it can converge to different solutions than classical methods. This is not a drawback but rather desirable, since our ultimate goal is minimizing the supervised loss rather than the physics-informed loss. To illustrate, recall that the fixed points of Gradient Descent and Gauss-Newton coincide, which follows from the rules for the Moore–Penrose inverse:

$$\theta^*_{GD} = (F^T F)^{-1} F^T c = F^{-1} c = (F^{-1} F)^{-1} F^{-1} c = \theta^*_{GN}. \tag{13}$$

In contrast, $\theta^*_{PS} = (BF)^{-1} Bc$, where $B$ represents a different, decoupled feedback propagation, so in general $\theta^*_{PS} \neq \theta^*_{GD}$. To better understand how these fixed points perform on the supervised loss, we next present two theorems. The first provides a generic relation between the two losses that holds for any $\theta$, not only fixed points.

**Theorem 4: Mutual bounds between physics-informed and supervised losses**

*The physics-informed loss $L_{PI}(n_i, \alpha)$ and supervised loss $L_{SV}(n_i)$, with $n_i$ the number of time steps and $\alpha$ the weight of the initial condition, bound each other as follows:*

$$L_{SV}(n_i) \leq \max(1/\alpha, 1) \left\| M^{-1} \right\|^2 \mu(n_i, \|M^{-1}\| \|N\|)^2 (n_i + 1) L_{PI}(n_i, \alpha),$$
$$L_{PI}(n_i, \alpha) \leq \max(\alpha, 1) \left( \|M\| + \max(\|N\|, 1) \right)^2 (n_i + 1) L_{SV}(n_i). \tag{14}$$

*Here $\mu$ is given by*

$$\mu(n, \lambda) = \begin{cases} \frac{1}{1-\lambda}, & \lambda < 1, \\ n + 1, & \lambda = 1, \\ \frac{\lambda^{n+1}}{\lambda - 1}, & \lambda > 1. \end{cases} \tag{15}$$

As a side note, the theorem uses the linearity of the time evolution scheme only through bounds on the operator norms $M$ and $N$. Hence, it extends to nonlinear physics provided these operators and their inverses are globally Lipschitz.

Notably, the supervised loss is bounded via the physics-informed loss. Unlike the second inequality, the first depends on the number of time steps $n_i$, so the physics-informed loss may exceed the

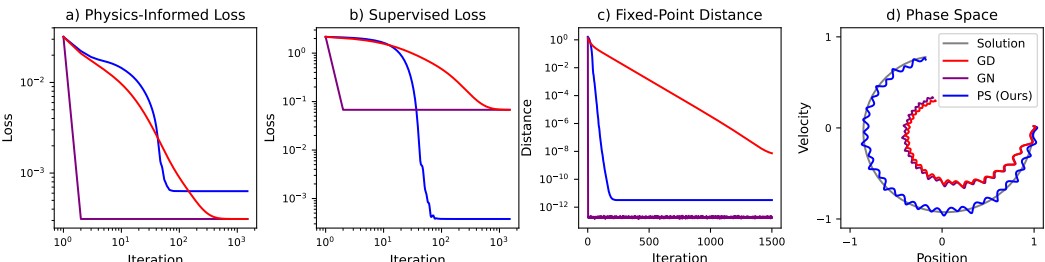

Figure 2: Harmonic oscillator: comparison of Gradient Descent (GD), Gauss-Newton (GN) and PitStop (PS). a) physics-informed loss, b) supervised loss, c) distance to the respective fixed-point, d) trajectories in phase space after 700 optimization steps.

supervised loss by orders of magnitude, scaling even exponentially if $\lambda > 1$. Thus, minimizing the physics-informed loss alone is questionable, since the supervised loss may remain large. An important exception occurs when the function approximator is expressive enough to fit the target perfectly. In this case, all three methods converge to the same solution, which is also a minimum of the supervised loss.

**Theorem 5: Physics-informed training under non-compact function approximation**

*If a perfect solution $\theta^*$ exists with $L_{PI} = 0$, then all fixed points coincide, $\theta^* = \theta^*_{GD} = \theta^*_{GN} = \theta^*_{PS}$, and this solution also satisfies $L_{SV} = 0$.*

Even when a perfect solution exists, PitStop surpasses Gradient Descent by converging faster. More importantly, in the realistic case of approximate solutions—which underlies the scalability of machine learning—its guaranteed convergence rate frees architectural choices to target alignment with the supervised optimum rather than correcting for Gradient Descent's inefficiency. This concludes our theoretical framework, which offers a comprehensive introduction to how classical optimization methods differ from PitStop's modified feedback backpropagation.

## 5 EXPERIMENTS

We present four experiments. The first is a toy problem illustrating concepts from the previous section. The second tests our theoretical predictions on a linear system. The third and fourth extend the analysis to nonlinear systems. Each setup follows the loss formulation described in Section 2, meaning we use function approximation to represent directly the physical fields $y(t, x)$ at fixed grid points. We discretize time derivatives of a given initial value problem to formulate a physics-informed loss $L_{PI}$ from the residuals, which we use to compute updates, and then we evaluate how closely our approximation resembles the true evolution of the system with a supervised $L_{SV}$. We compare several optimization methods to solve this task, tuning each individually for best performance. Additional details about the setups are in the appendix.

**Toy example** In the first example, we define a linear system with three equations and two unknowns, corresponding to a physics-informed setup with one initial condition, two time steps, and two parameters. The setup is overdetermined, satisfying Theorems 1–4. Figure 1 illustrates the scenario: the physics-informed loss (a) shows the expected elliptic contours, with the GD/GN fixed-point at the center, PitStop's fixed-point marked in blue, and the supervised optimum in green. The supervised loss landscape (b) is also quadratic, with the green dot optimal, GD/GN is far off, and PitStop closer, and all three points satisfy the inequalities from Theorem 4 ($L_{SV} \le 300 \cdot L_{PI}$, $L_{PI} \le 14.6 \cdot L_{SV}$). Part (c) shows GD update directions following the valley-like regions typical of ill-conditioned landscapes. In contrast (d), PitStop exhibits spiraling streamlines from the non-symmetry of the fixed-point operator, intuitively enabling escape from slow valleys and accelerating optimization.

**Harmonic oscillator** Next, we test our theory on the harmonic oscillator, a fundamental linear differential equation that is more ill-conditioned than the toy example, making the challenges of

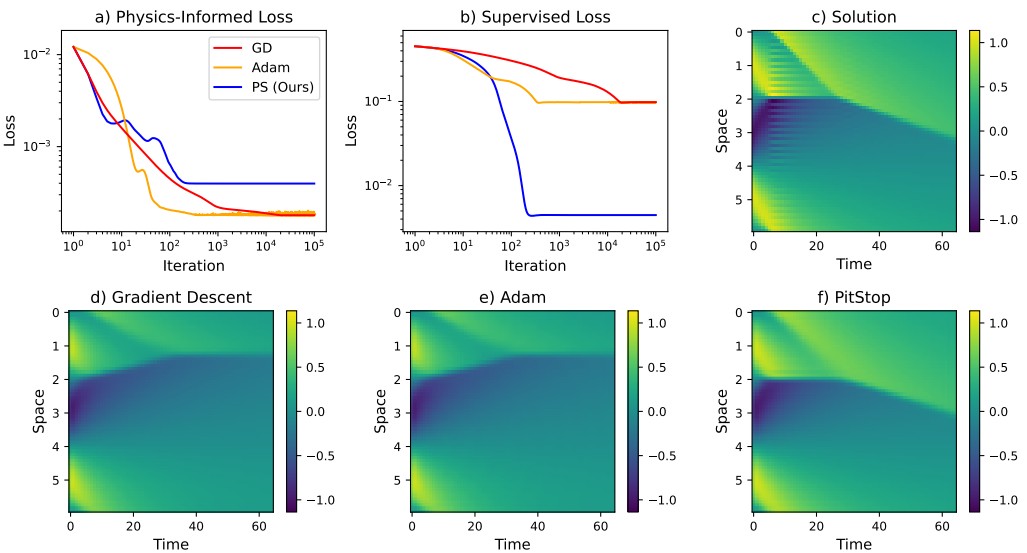

Figure 3: Burgers' equation: comparison of Gradient Descent (GD), Adam and PitStop (PS).
a) physics-informed loss, b) supervised loss, c) numerical solution d)-f) fixed-points.

physics-informed training apparent. Figure 2 shows the results: the supervised loss can now be orders of magnitude higher, consistent with Theorem 4 inequalities ($L_{SV} \leq 1.85 \cdot 10^6 \cdot L_{PI}$ and $L_{PI} \leq 1.84 \cdot 10^2 \cdot L_{SV}$). PitStop reaches a better fixed point with a supervised loss two orders of magnitude lower than GD/GN. Part (d) visualizes the learned trajectories: classical methods minimize the physics-informed loss by transitioning toward a zero trajectory over time, whereas PitStop approaches the true physical solution. Convergence speed, measured as distance to each method's fixed point in (c), shows GN solves in one iteration at the cost of inversion, while PitStop clearly converges faster than GD, in line with Theorem 3.

**Burgers' equation**    In the next experiment, we study Burgers' equation, a nonlinear PDE with one spatial and one temporal dimension (Figure 3). Due to the higher dimensionality, we replace Gauss-Newton with Adam, which mitigates ill-conditioning without full matrix inversions. As with the linear system, classical methods and PitStop reach different fixed points, PitStop converges faster (Theorem 3), and supervised losses far exceed physics-informed losses (Theorem 4). Importantly, only PitStop accurately predicts the solution's long-term behavior.

**Navier-Stokes equations**    As the final experiment, we study lid-driven cavity flow governed by the Navier–Stokes equations with two spatial and one temporal dimension (Figure 4). The behavior matches the other experiments. As before, there are clear differences between the converged classical and PitStop fixed points as well as how fast these points are found. This demonstrates that even on challenging fluid dynamics systems, the phenomena described in our theoretical analysis are observed, and our method remains effective.

## 6    RELATED WORK

Our work is inspired by studies on non-gradient methods for value estimation in Reinforcement Learning Sutton (1988); Tsitsiklis & Van Roy (1997); Schnell et al. (2025). In particular, we generalized the convergence proofs to non-scalar fields for PitStop, making it suitable for physical systems. Notably, the interest within Reinforcement Learning in non-gradient methods is based on empirically observed performance improvements, whereas our paper provides the most comprehensive theoretical framework to justify these methods to date.

Most works on physics-informed objectives are empirical in nature. A few theoretical studies adapt error bounds from the supervised learning literature to physics-informed settings Shin et al. (2020);

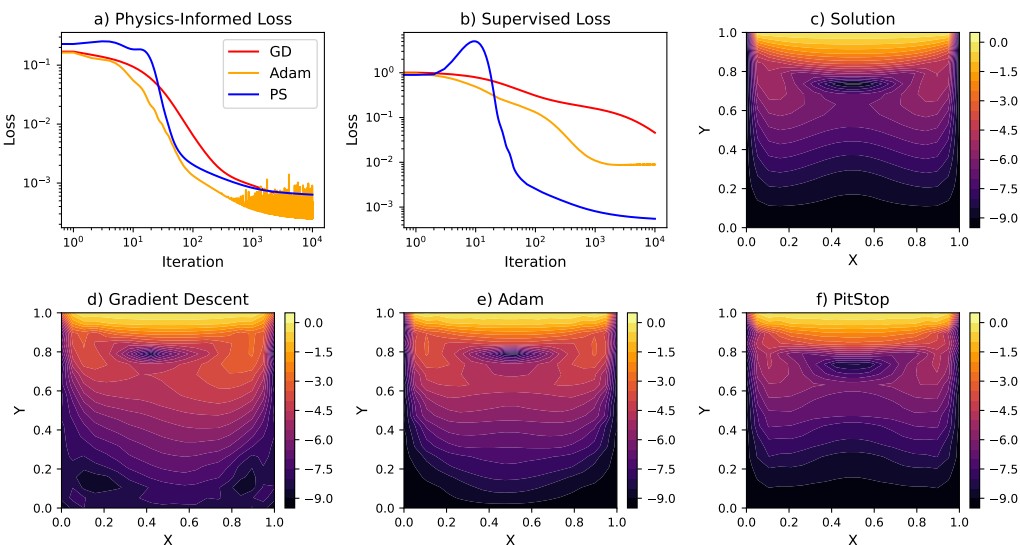

Figure 4: Navier–Stokes equation: comparison of Gradient Descent (GD), Adam, and PitStop (PS): (a) physics-informed loss, (b) supervised loss, (c) numerical solution, (d–f) fixed points showing velocity vector length at the final time step (log scale)

Wu et al. (2022); Mishra & Molinaro (2023); Shin et al. (2023), where the generalization error refers to the discrepancy outside the collocation points used during training. Further works have derived bounds for specific scenarios, like linear parabolic De Ryck & Mishra (2022) and hyperbolic Qian et al. (2023) partial differential equations. Various solutions to the optimization challenges of physics-informed training have been proposed, such as higher-order optimization methods Kiyani et al. (2025), reweighting loss terms McClenny & Braga-Neto (2023), regularization strategies Doumèche et al. (2023), or improved initialization strategies Liu et al. (2022). All of them have in common that they work within the classical picture of optimization, the inherent shortcomings of which we have explained in our work.

# 7 CONCLUSION

Our paper introduces a theoretical framework for optimizing a target loss by computing updates based on a different loss. This naturally aligns with a physics-informed learning task, where a supervised loss is minimized through updates derived from a physics-informed loss. Despite being unintuitive at first, this methodology works because an ideal solution minimizes both losses simultaneously.

What sets our work apart is its full embrace of the approximate nature of machine learning in this context, leading us to conclude that abandoning the classical optimization methods is a necessity: As we demonstrated with our PitStop method, convergence can be guaranteed through other mechanisms than steepest descent steps, the convergence rate is much less affected by ill-conditioning, and its fixed-point has no drawbacks compared to classical methods. This stems from the fact that the least-squared property becomes irrelevant when minimizing another loss.

Future work should explore how the fixed-point depends on the function approximator and how our framework can be extended to nonlinear situations. It is also essential to investigate whether our method can be generalized to continuous time derivative formulations. If not, our work would expose a fundamental weakness of continuous-time physics-informed setups that compute physical derivatives via automatic differentiation.

## REPRODUCABILITY STATEMENT

To ensure reproducibility of our work we will publish the source code for our experiments upon acceptance. Details on our experiments are also provided in the appendix.

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

# Appendix

## A  MATHEMATICAL PROOFS

### A.1  THEOREM 1

**Statement of Theorem 1**

*In the linear case, PitStop performs updates of the form $u_{PS} = B(F\theta - c)$, where:*

$$
\begin{aligned}
F = TA \quad &\text{with} \quad T = \sqrt{\alpha} \cdot E \otimes 1 + (1 - E) \otimes M - O \otimes N \\
B = A^\dagger U^\dagger \quad &\text{with} \quad U = \sqrt{\alpha} \cdot E \otimes 1 + (1 - E) \otimes M
\end{aligned}
\tag{16}
$$

*Here, $A$ denotes the row-stacked constant Jacobians $A_i = \frac{\partial y}{\partial \theta}\big|_{t_i}$, i.e. $A^\dagger = [A_0^\dagger \quad A_1^\dagger \quad \dots \quad A_{n_i}^\dagger]$. Furthermore, $E \in \mathbb{R}^{n_j \otimes n_j}$ with $E_{ab} = 1$ if $a = b = 0$ and $E_{ab} = 0$ elsewhere, and $O \in \mathbb{R}^{n_j \times n_j}$ with $O_{ab} = 1$ if $a - b = 1$ and $O_{ab} = 0$ elsewhere, and $1$ are identity matrices of appropriate size.*

**Proof of Theorem 1**

The initial condition takes the form:

$$
R_0 = y_0 - y_0^{sol}
\tag{17}
$$

The linearity assumption on differential equation leads to the residual:

$$
R_i = M y_i - N y_{i-1} - c_i
\tag{18}
$$

Using linearity of the function approximator simplifies these two expressions to:

$$
\begin{aligned}
R_0 &= A_0 \theta - c_0 \\
R_i &= M A_i \theta - N A_{i-1} \theta - c_i
\end{aligned}
\tag{19}
$$

Here, we introduced the constant matrices $A_i = \frac{\partial y}{\partial \theta}\big|_{t_i}$. This allows us to rewrite the loss:

$$
\begin{aligned}
L_{PI} &= \alpha \|R_0\|^2 + \sum_{i=1}^{n_i} \|R_i\|^2 \\
&= \alpha \|A_0 \theta - c_0\|^2 + \sum_{i=1}^{n_i} \|M A_i \theta - N A_{i-1} \theta - c_i\|^2
\end{aligned}
\tag{20}
$$

Comparing terms with $L_{PI} = \|F\theta - c\|^2$, we find that $F = TA$, where $A$ denotes the row-stacked Jacobians, i.e. $A^\dagger = [A_0^\dagger \quad A_1^\dagger \quad \dots \quad A_{n_i}^\dagger]$, and $T$ is given by:

$$
T = \sqrt{\alpha} \cdot E \otimes 1 + (1 - E) \otimes M - O \otimes N
\tag{21}
$$

Here, $E \in \mathbb{R}^{n_j \otimes n_j}$ with $E_{ab} = 1$ if $a = b = 0$ and $E_{ab} = 0$ elsewhere, and $O \in \mathbb{R}^{n_j \otimes n_j}$ with $O_{ab} = 1$ if $a - b = 1$ and $O_{ab} = 0$ elsewhere.

Next, we transform the update equation:

$$
\begin{aligned}
u_{PS} &= \alpha \cdot \left(\frac{dy_0}{d\theta}\right)^\dagger \cdot R_0 + \sum_{i=1}^{n_i} \left(M \frac{dy_i}{d\theta}\right)^\dagger \cdot R_i \\
&= \alpha \cdot A_0^\dagger \cdot R_0 + \sum_{i=1}^{n_i} A_i^\dagger M^\dagger R_i
\end{aligned}
\tag{22}
$$

To receive $B$, we compare this to the definition $u_{PS} = B(F\theta - c)$ and find $B = A^\dagger U^\dagger$ with:

$$
U = \sqrt{\alpha} \cdot E \otimes 1 + (1 - E) \otimes M
\tag{23}
$$

## A.2 THEOREM 2

**Definition (Block Diagonal Dominance)** Let $W \in \mathbb{C}^{nk \times nk}$ be partitioned into blocks $W^{(i,j)} \in \mathbb{C}^{k \times k}$. $W$ is strictly block-diagonally dominant if

$$\left\| (W^{(i,i)})^{-1} \right\|^{-1} > \sum_{j \neq i} \left\| W^{(i,j)} \right\| \quad \forall i. \tag{24}$$

**Lemma (Block Gershgorin).** Each eigenvalue $\lambda$ of $W$ satisfies, for some $i$,

$$\left\| (W^{(i,i)} - \lambda)^{-1} \right\|^{-1} \leq \sum_{j \neq i} \left\| W^{(i,j)} \right\|. \tag{25}$$

**Proof of Lemma**

The definition and the lemma were introduced and proven in Feingold & Varga (1962) (Theorem 1 and 2, Equations 2.1 to 2.5).

**Corollary** If $W$ is symmetric, $W^{(i,i)} \succ 0$ for all $i$, and $W$ is strictly block-diagonally dominant, then $W \succ 0$.

**Proof of Corollary** Let $\mu$ be any eigenvalue of $W$, and choose $i$ satisfying the lemma. If $\mu \in \text{spec}(W^{(i,i)})$, then $\mu > 0$. Otherwise,

$$\sigma_{\min}(W^{(i,i)} - \mu) \leq \sum_{j \neq i} \left\| W^{(i,j)} \right\|. \tag{26}$$

Since $W^{(i,i)} \succ 0$, we have $\sigma_{\min}(W^{(i,i)} - \mu) \geq \lambda_{\min}(W^{(i,i)}) - \mu$, giving

$$\mu \geq \lambda_{\min}(W^{(i,i)}) - \sum_{j \neq i} \left\| W^{(i,j)} \right\| > 0. \tag{27}$$

Hence all eigenvalues of $W$ are positive, so $W \succ 0$.

**Statement of Theorem 2**

*Let $M$ be non-singular, and let $M$ and $N$ satisfy $\sigma_{\min}(M^\dagger M) > \sigma_{\max}(M^\dagger N)$, i.e., the smallest singular value of $M^\dagger M$ is strictly greater than the largest singular value of $M^\dagger N$. If the weight factor $\alpha$ for the initial condition term is chosen such that $\alpha > \sigma_{\max}(M^\dagger N)$, then PitStop converges.*

**Proof of Theorem 2**

From Theorem 1, we have:

$$\begin{aligned} F &= TA \quad \text{with} \quad T = \sqrt{\alpha} \cdot E \otimes 1 + (1 - E) \otimes M - O \otimes N \\ B &= A^\dagger U^\dagger \quad \text{with} \quad U = \sqrt{\alpha} \cdot E \otimes 1 + (1 - E) \otimes M \end{aligned} \tag{28}$$

It follows that $BF = A^\dagger U^\dagger TA$ with:

$$U^\dagger T = \alpha \cdot E \otimes 1 + (1 - E) \otimes M^\dagger M - O \otimes M^\dagger N \tag{29}$$

To show how the properties of $U^\dagger T$ determine convergence of PitStop, we need to generalize positive-definiteness to nonsymmetric matrices:

A real matrix $K$ is defined to be (strictly) spectrally positive definite (SPD) if the real part of all its eigenvalues is (strictly) positive. Similarly, $K$ is (strictly) geometrically positive definite (GPD) if $\langle x, Kx \rangle$ is (strictly) positive for all nonzero x. These abbreviations, SPD and GPD, are used in the following.

The argument proceeds as follows: Assume $U^\dagger T$ is strictly GPD. Since GPD is preserved under congruence transformations, $A^\dagger U^\dagger TA$ is strictly GPD. GPD implies SPD, so $A^\dagger U^\dagger TA$ is also strictly SPD. Consequently, the iteration operator $1 - \eta A^\dagger U^\dagger TA$ is contractive for sufficiently small learning rates $\eta$.

Consequently, all we need to do is show that $U^\dagger T$ is GPD which is equivalent to the symmetric part $U^\dagger T + T^\dagger U$ being positive definite.

$$U^\dagger T + T^\dagger U = 2\alpha \cdot E \otimes 1 + 2 \cdot (1 - E) \otimes M^\dagger M - O \otimes M^\dagger N - O^\dagger \otimes N^\dagger M \qquad (30)$$

For this, we use the corollary on $U^\dagger T + T^\dagger U$. The first criterion, symmetry, is given. The second criterion, strict positive definiteness of the diagonal blocks, means $2\alpha \cdot 1 > 0$, which is trivially true, and $M^\dagger M$ is strictly positive definite if $M$ is non-singular. The third condition, being strictly block-diagonally dominant, simplifies as $U^\dagger T + T^\dagger U$ is block-tridiagonal matrix. Note that $\|M^\dagger N\| = \|N^\dagger M\|$. For the first row, we require that $\alpha > \|M^\dagger N\|$ and the other rows lead to the condition $\mathcal{L}(M^T M) > \|M^\dagger N\|$, which translates to $\sigma_{\min}(M^\dagger M) > \sigma_{\max}(M^\dagger N)$ for the Euclidean norm. Fulfilling all of these criteria yields $U^\dagger T + T^\dagger U > 0$.

### A.3 THEOREM 3

**Statement:**

*Assume PitStop converges. Define the condition number $\kappa^X(A) = \lambda_{\max}(A^\dagger X A)/\lambda_{\min}(A^\dagger X A)$ for $X$ full rank and $A$ the feature matrix. Assume a distribution $p(A)$ over relevant feature matrices $A$ and a model for condition number fulfilling $p(\kappa^X|A) = p(\kappa^X)$ when $X$ is symmetric. Then, PitStop has a lower condition number than Gradient Descent: $\mathbb{E}[\kappa^{U^\dagger T}] \leq \mathbb{E}[\kappa^{T^\dagger T}]$*

**Proof:**

The proof relies on Theorem 2 in Schnell et al. (2025), which states that strictly positive-definite, non-symmetric matrices have a more balanced eigenspectrum. Therefore, we can prove our claim by proving that PitStop is strictly positive-definite and non-symmetric.

By assuming of convergence of PitStop, we already have that the fixed-point operator is strictly positive definite. From our Theorem 1, we have for the fixed-point operator of PitStop:

$$U^\dagger T = \alpha \cdot E \otimes 1 + (1 - E) \otimes M^\dagger M - O \otimes M^\dagger N \qquad (31)$$

We find then for the antisymmetric component:

$$U^\dagger T - T^\dagger U = -O \otimes M^\dagger N + O^\dagger \otimes N^\dagger M \qquad (32)$$

This is non-zero because $M$ and $N$ are matrices encoding a time stepping scheme.

Then, we can estimate the condition numbers in expectation:

$$\mathbb{E}[\kappa^{U^\dagger T}] = \int \kappa^{U^\dagger T}(A)dp(A) \leq \int \kappa^{U^\dagger T + T^\dagger U}(A)dp(A) = \int \kappa^{T^\dagger T}(A)dp(A) = \mathbb{E}[\kappa^{T^\dagger T}] \quad (33)$$

The inequality is an application of the mentioned Theorem 2 in Schnell et al. (2025) and the equality afterwards due to the condition number model of the assumptions of this theorem.

### A.4 THEOREM 4

**Statement**

*The physics-informed loss $L_{PI}(n_i, \alpha)$ and supervised loss $L_{SV}(n_i)$, with $n_i$ the number of time steps and $\alpha$ the weight of the initial condition, bound each other as follows:*

$$L_{SV}(n_i) \leq \max(1/\alpha, 1) \|M^{-1}\|^2 \mu(n_i, \|M^{-1}\|\|N\|)^2 (n_i + 1) L_{PI}(n_i, \alpha),$$
$$L_{PI}(n_i, \alpha) \leq \max(\alpha, 1) (\|M\| + \max(\|N\|, 1))^2 (n_i + 1) L_{SV}(n_i). \qquad (34)$$

*Here $\mu$ is given by*

$$\mu(n, \lambda) = \begin{cases} \frac{1}{1-\lambda}, & \lambda < 1, \\ n+1, & \lambda = 1, \\ \frac{\lambda^{n+1}}{\lambda-1}, & \lambda > 1. \end{cases} \tag{35}$$

**Proof**

We will prove:

$$L_{SV}(n_i) \leq \left\| M^{-1} \right\|^2 \cdot \mu(n_i, \left\| M^{-1} \right\| \cdot \|N\|)^2 \cdot (n_i + 1) \cdot L_{PI}(n_i, 1)$$
$$L_{PI}(n_i, 1) \leq \left( \|M\| + \max \left( \|N\|, 1 \right) \right)^2 \cdot (n_i + 1) \cdot L_{SV}(n_i) \tag{36}$$

Then the claim follows simply from:

$$L_{PI}(n_i, \alpha) \leq \max(\alpha, 1) \cdot L_{PI}(n_i, 1)$$
$$L_{PI}(n_i, 1) \leq \max(1/\alpha, 1) \cdot L_{PI}(n_i, \alpha) \tag{37}$$

Let us introduce $sv(i) = \left\| y_i - y_i^{\text{sol}} \right\|$ and $pi(i) = \|R_i\|$ for $i \geq 0$. In particular, $sv(0) = pi(0)$. Then we have $L_{SV}(n_i) = \sum_{i=0}^{n_i} sv(i)^2$ and $L_{PI}(n_i, 1) = \sum_{i=0}^{n_i} pi(i)^2$.

We can estimate a supervised term through the corresponding physics-informed term and the previous supervised term, for $i > 0$:

$$\begin{aligned} sv(i) &= \left\| y_i - y_i^{\text{sol}} \right\| \\ &\leq \left\| M^{-1} \right\| \cdot \left\| My_i - My_i^{\text{sol}} \right\| \\ &\leq \left\| M^{-1} \right\| \cdot \left( \left\| My_i - Ny_{i-1} - c_i \right\| + \left\| Ny_{i-1} + c_i - My_i^{\text{sol}} \right\| \right) \\ &= \left\| M^{-1} \right\| \cdot \left( \left\| My_i - Ny_{i-1} - c_i \right\| + \left\| Ny_{i-1} + c_i - Ny_{i-1}^{\text{sol}} - c_i \right\| \right) \\ &= \left\| M^{-1} \right\| \cdot \left( \left\| My_i - Ny_{i-1} - c_i \right\| + \left\| N(y_{i-1} - y_{i-1}^{\text{sol}}) \right\| \right) \\ &\leq \left\| M^{-1} \right\| \cdot \left( \left\| My_i - Ny_{i-1} - c_i \right\| + \|N\| \cdot \left\| (y_{i-1} - y_{i-1}^{\text{sol}}) \right\| \right) \\ &\leq \left\| M^{-1} \right\| \cdot pi(i) + \left\| M^{-1} \right\| \cdot \|N\| \cdot sv(i-1) \end{aligned} \tag{38}$$

Induction leads to an expression of the supervised term through only physics-informed terms. Note that the $k = 0$-term has been overestimated by a factor of $\left\| M^{-1} \right\|$ for simplicity.

$$sv(i) \leq \sum_{k=0}^{i} \left\| M^{-1} \right\|^{i-k+1} \cdot \|N\|^{i-k} \cdot pi(k) \tag{39}$$

With that, we can estimate the complete supervised loss:

$$
\begin{aligned}
L_{SV}(n_i) &= \sum_{i=0}^{n_i} sv(i)^2 \\
&\leq \left( \sum_{i=0}^{n_i} sv(i) \right)^2 \\
&\leq \left( \sum_{i=0}^{n_i} \sum_{k=0}^{i} \left\| M^{-1} \right\|^{i-k+1} \cdot \left\| N \right\|^{i-k} \cdot pi(k) \right)^2 \\
&= \left\| M^{-1} \right\|^2 \cdot \left( \sum_{i=0}^{n_i} \sum_{k=0}^{i} \left( \left\| M^{-1} \right\| \cdot \left\| N \right\| \right)^{i-k} \cdot pi(k) \right)^2 \\
&\leq \left\| M^{-1} \right\|^2 \cdot \left( \sum_{k=0}^{n_i} \left( \left\| M^{-1} \right\| \cdot \left\| N \right\| \right)^{k} \right)^2 \cdot \left( \sum_{k=0}^{n_i} pi(k) \right)^2 \\
&\leq \left\| M^{-1} \right\|^2 \cdot \mu(n_i, \left\| M^{-1} \right\| \cdot \left\| N \right\|)^2 \cdot (n_i+1) \cdot \left( \sum_{k=0}^{n_i} pi(k)^2 \right) \\
&\leq \left\| M^{-1} \right\|^2 \cdot \mu(n_i, \left\| M^{-1} \right\| \cdot \left\| N \right\|)^2 \cdot (n_i+1) \cdot L_{PI}(n_i, 1)
\end{aligned}
\tag{40}
$$

The last step applies the common estimates for geometric series. This finishes the proof of the first inequality. For the second one, we start with:

$$
\begin{aligned}
pi(i) &= \left\| M y_i - N y_{i-1} - c_i \right\| \\
&\leq \left\| M y_i - M y_i^{\text{sol}} \right\| + \left\| M y_i^{\text{sol}} - N y_{i-1} - c_i \right\| \\
&\leq \left\| M \right\| \cdot \left\| y_i - y_i^{\text{sol}} \right\| + \left\| N y_{i-1}^{\text{sol}} + c_i - N y_{i-1} - c_i \right\| \\
&\leq \left\| M \right\| \cdot \left\| y_i - y_i^{\text{sol}} \right\| + \left\| N y_{i-1}^{\text{sol}} - N y_{i-1} \right\| \\
&\leq \left\| M \right\| \cdot \left\| y_i - y_i^{\text{sol}} \right\| + \left\| N \right\| \cdot \left\| y_{i-1}^{\text{sol}} - y_{i-1} \right\| \\
&\leq \left\| M \right\| \cdot sv(i) + \left\| N \right\| \cdot sv(i-1)
\end{aligned}
\tag{41}
$$

For the full physics-informed loss, we find the second inequality:

$$
\begin{aligned}
L_{PI}(n_i, 1) &= \sum_{i=0}^{n_i} pi(i)^2 \\
&\leq \left( \sum_{i=0}^{n_i} pi(i) \right)^2 \\
&\leq \left( \left\| M \right\| sv(n_i) + \left( \left\| M \right\| + \left\| N \right\| \right) \sum_{i=1}^{n_i-1} sv(i) + \left( \left\| 1 \right\| + \left\| N \right\| \right) sv(0) \right)^2 \\
&\leq \left( \left\| M \right\| + \max\left( \left\| N \right\|, 1 \right) \right)^2 \cdot \left( \sum_{i=0}^{n_i} sv(i) \right)^2 \\
&\leq \left( \left\| M \right\| + \max\left( \left\| N \right\|, 1 \right) \right)^2 \cdot (n_i+1) \cdot \left( \sum_{i=0}^{n_i} sv(i)^2 \right) \\
&\leq \left( \left\| M \right\| + \max\left( \left\| N \right\|, 1 \right) \right)^2 \cdot (n_i+1) \cdot L_{SV}(n_i)
\end{aligned}
\tag{42}
$$

## A.5 THEOREM 5

**Statement**

*If a perfect solution $\theta^*$ exists with $L_{PI} = 0$, then all fixed points coincide, $\theta^* = \theta_{GD}^* = \theta_{GN}^* = \theta_{PS}^*$, and this solution also satisfies $L_{SV} = 0$.*

**Proof**

Gradient Descent and Gauss-Newton approach $\theta^*$ for two reasons: First they approach the least-squares solution in a quadratic objective. Secondly, our analysis assumed the requirements of Banach's fixed point theorem with a contracting fixed point operator, which states the fixed point is unique. Therefore, $\theta^* = \theta_{GD}^* = \theta_{GN}^*$.

Again by the uniqueness property, it follows that at the point $\theta^*$ the update of PitStop vanishes, because of its factor $F\theta - c$, which is 0 since $L_{PI}$ is 0. Hence, $\theta^*$ is also the fixed point of PitStop, $\theta^* = \theta_{PS}^*$.

From our Theorem 4, it follows directly that if $L_{PI} = 0$, then $L_{SV} = 0$.

## B  Experimental details

### B.1  Toy example

Following our notation in the main part, our toy example is given by:

$$M = 1.1 \quad N = 1 \quad \alpha = 1.1 \quad c = \begin{pmatrix} 2 \\ -2 \\ -2 \end{pmatrix} \tag{43}$$

This leads to:

$$T = \begin{pmatrix} 1.1 & 0 & 0 \\ 1 & 1.1 & 0 \\ 0 & 1 & 1.1 \end{pmatrix} \quad U = \begin{pmatrix} 1.1 & 0 & 0 \\ 0 & 1.1 & 0 \\ 0 & 0 & 1.1 \end{pmatrix} \tag{44}$$

As function approximator, we use:

$$A = \begin{pmatrix} 6 & 0 \\ -3 & 4 \\ 2 & -2 \end{pmatrix} \tag{45}$$

### B.2  Harmonic oscillator

Figure 5 further illustrates the optimization process over the iteration steps.

The harmonic oscillator is a fundamental linear second-order differential equations. Its higher-dimensional first-order form is given by:

$$\frac{d}{dt} \begin{pmatrix} y^{\text{pos}} \\ y^{\text{vel}} \end{pmatrix} = P \cdot \begin{pmatrix} y^{\text{pos}} \\ y^{\text{vel}} \end{pmatrix} = \begin{pmatrix} 0 & 1 \\ -1 & 0 \end{pmatrix} \cdot \begin{pmatrix} y^{\text{pos}} \\ y^{\text{vel}} \end{pmatrix} \quad \text{with} \quad \begin{pmatrix} y_0^{\text{pos}} \\ y_0^{\text{vel}} \end{pmatrix} = \begin{pmatrix} 1 \\ 0 \end{pmatrix} \tag{46}$$

Applying the implicit Euler scheme with a time step of $\tau = 0.1$, we set $M = 1 + \tau P$ and $N = 1$, and additionally set the weight factor of the initial condition $\alpha = 1 + \tau^2$ to fulfill the requirements of Theorem 2 for convergence of our method PitStop. We evolve this system for $45$ time steps and parametrize each component of the state vector by a linear function approximator with $30$ parameters. The features are localized Gaussian functions. The optimizers are Gradient Descent, Gauss-Newton, and PitStop. The learning rates were individually tuned and are close to optimal (GD: 50, GN: 1, PS 25). For example, a learning rate of 55 was too large and led to divergence.

### B.3  Burgers' equation

The Burgers' equation is given by:

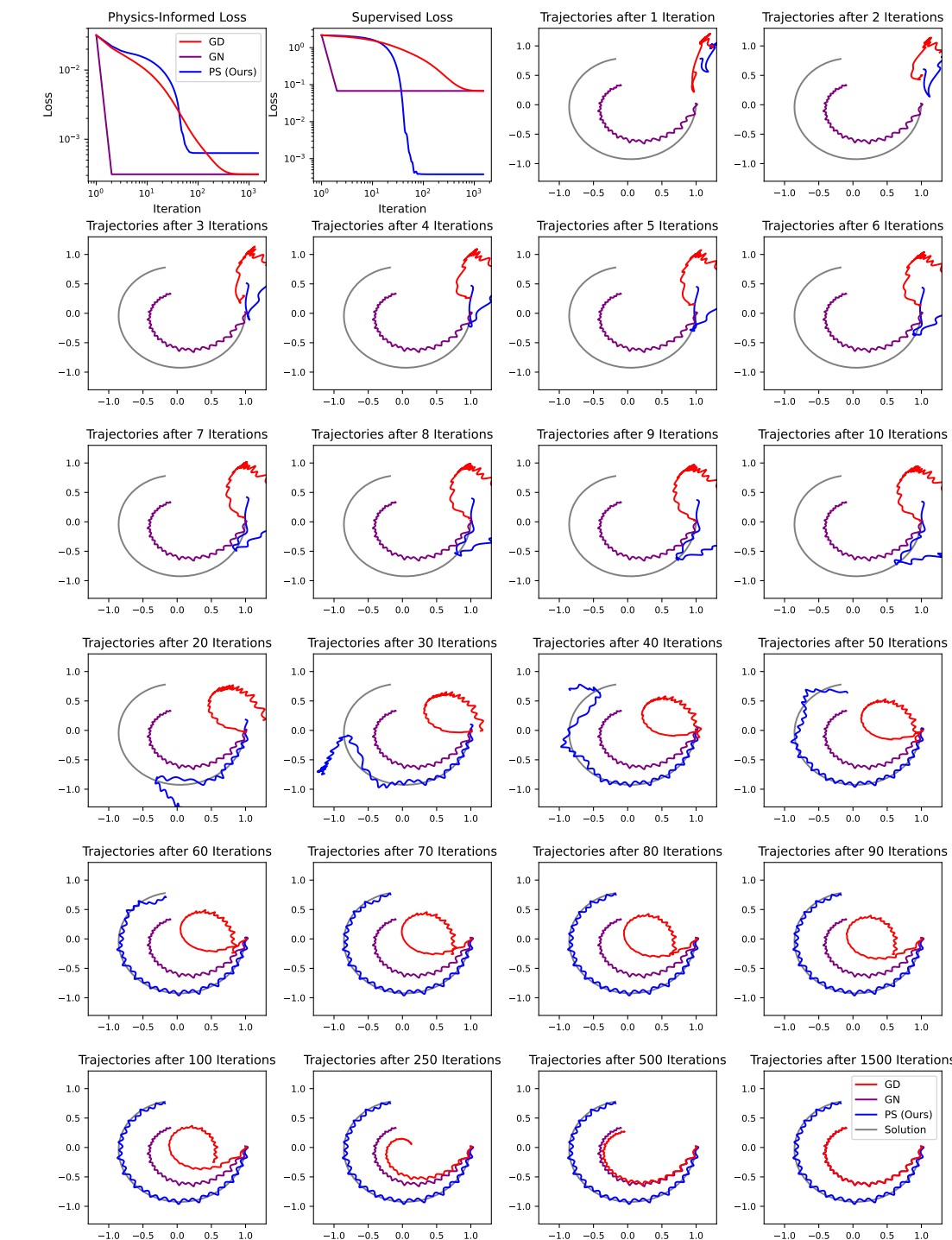

Figure 5: Harmonic oscillator: illustration of the optimization process over iteration steps.

$$\partial_t y = 0.1 \cdot \partial_x^2 y - y \cdot \partial_x y \tag{47}$$

We use periodic boundary conditions and our initial condition is $\sin(\pi \cdot x/2)$.

The spatial domain is $[0, 6]$. We use $64$ equidistant collocation points. To evolve the system for a time step of $\tau = 1$, we use 10 explicit Euler steps with a step of $0.1$ and compute spatial derivatives in Fourier space. We use a weight factor of 1 on the initial condition. We evolve for $64$ time steps. We approximate the solution by a linear function approximator with 2500 parameters. The features are two-dimensional Gaussian functions. We optimize with Gradient Descent, Adam and PitStop. The learning rates were individually tuned and are close to optimal (GD: 100, Adam: 0.02, PS 100).

### B.4 NAVIER-STOKES EQUATIONS

We consider the two-dimensional, incompressible Navier–Stokes equations in velocity–pressure formulation:

$$\frac{\partial u}{\partial t} + u\frac{\partial u}{\partial x} + v\frac{\partial u}{\partial y} = -\frac{1}{\rho}\frac{\partial p}{\partial x} + \nu\left(\frac{\partial^2 u}{\partial x^2} + \frac{\partial^2 u}{\partial y^2}\right), \tag{48}$$

$$\frac{\partial v}{\partial t} + u\frac{\partial v}{\partial x} + v\frac{\partial v}{\partial y} = -\frac{1}{\rho}\frac{\partial p}{\partial y} + \nu\left(\frac{\partial^2 v}{\partial x^2} + \frac{\partial^2 v}{\partial y^2}\right), \tag{49}$$

$$\frac{\partial u}{\partial x} + \frac{\partial v}{\partial y} = 0, \tag{50}$$

where $u(x, y, t)$ and $v(x, y, t)$ are the velocity components in the $x$- and $y$-directions, $p(x, y, t)$ is the pressure, $\rho$ is the fluid density, and $\nu$ is the kinematic viscosity.

The domain is discretized on a uniform Cartesian grid of size $(n_x, n_y)$ with spacing $\Delta x$ and $\Delta y$. Velocities and pressure are collocated at the same grid points. Temporal advancement uses a forward Euler scheme with time step $\Delta t$.

Spatial derivatives are approximated by second-order central differences. For example,

$$\left.\frac{\partial u}{\partial x}\right|_{i,j} \approx \frac{u_{i,j+1} - u_{i,j-1}}{2\Delta x},$$

$$\left.\frac{\partial^2 u}{\partial x^2}\right|_{i,j} \approx \frac{u_{i,j+1} - 2u_{i,j} + u_{i,j-1}}{\Delta x^2},$$

with analogous formulas in the $y$-direction.

At each time step, the intermediate velocity field is projected onto a divergence-free space by solving the pressure Poisson equation

$$\frac{\partial^2 p}{\partial x^2} + \frac{\partial^2 p}{\partial y^2} = \rho\left(\frac{1}{\Delta t}\left(\frac{\partial u}{\partial x} + \frac{\partial v}{\partial y}\right) - \left(\frac{\partial u}{\partial x}\right)^2 - 2\frac{\partial u}{\partial y}\frac{\partial v}{\partial x} - \left(\frac{\partial v}{\partial y}\right)^2\right), \tag{51}$$

discretized again using second-order central differences. The equation is solved iteratively (Jacobi method) for a prescribed number of iterations $n_{\text{it}}$.

We impose no-slip conditions on all walls, except for the top lid, where a constant horizontal velocity $u = U_{\text{lid}}$ is prescribed and $v = 0$. Pressure boundary conditions are Neumann on vertical walls, Dirichlet at the top, and zero-gradient at the bottom.

The computational domain is a square cavity of size $L_x = L_y = 1.0$ discretized on a $20 \times 20$ uniform grid. The simulation is advanced for $n_t = 20$ time steps with a step size of $\Delta t = 0.005$. The pressure Poisson equation is iterated $n_{\text{it}} = 3$ times per time step. Fluid properties are set to density $\rho = 1.0$ and kinematic viscosity $\nu = 0.1$. The lid is driven with a constant velocity $U_{\text{lid}} = 1.0$ in the $x$-direction. We approximate the solution using a linear function approximator with 3375 parameters. The features are two-dimensional Gaussian functions. We optimize with Gradient Descent, Adam and PitStop. The learning rates were individually tuned and are close to optimal (GD: 6, Adam: 0.01, PS 50).

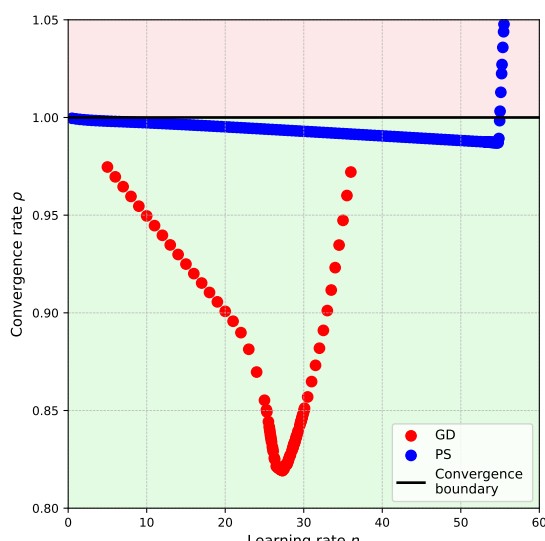

Figure 6: Convergence rate over learning rate for the harmonic oscillator for Gradient Descent (GD) and PitStop (PS). The algorithms converge if the convergence rate is less than $1$.

## C  ADDITIONAL EXPERIMENTS

### C.1  LEARNING RATE COMPARISON ON THE HARMONIC OSCILLATOR

To illustrate the effect of different learning rates, we plot the corresponding convergence rates for Gradient Descent and PitStop in Figure 6. For Gradient Descent, we observe a piecewise-linear dependence around the optimal learning rate, as expected for methods with symmetric fixed-point operators. In contrast, the curve for PitStop exhibits richer structure due to the non-symmetry of its fixed-point operator, which also allows for better convergence rates.

### C.2  STATISTICAL EVALUATION ON THE BURGERS' EQUATION

In this subsection, we list the results of further runs on the Burgers' equation with the same experimental setup as in the main part of the paper except for different initial conditions and initialization of the model parameters to explore the convergence behavior of PitStop further in nonlinear setups beyond the theoretical analysis in the main part. Figure 1, 2 and 3 show these results after 100, 1000, and 10000 iteration steps, respectively.

The results further strengthen the insights from the main part: For the established methods Gradient Descent and Adam, the supervised loss can be high even when the physics-informed loss is low. PitStop reaches its final configuration in less iteration steps. PitStop does not minimize the physics-informed loss as much as the other methods but its solution is more optimal in terms of the supervised loss.

### C.3  STATISTICAL EVALUATION ON THE NAVIER-STOKES EQUATIONS

In this subsection, we list the results of further runs on the Navier-Stokes equations with the same experimental setup as in the main part of the paper except for different initial conditions and initialization of the model parameters to explore the convergence behavior of PitStop further in nonlinear setups beyond the theoretical analysis in the main part. Figure 4, 5 and 6 show these results after 100, 1000, and 10000 iteration steps, respectively.

The results are similar to the Burgers' equation: For the established methods Gradient Descent and Adam, the supervised loss can be high even when the physics-informed loss is low. PitStop reaches

Table 1: Physics-informed (PI) and (SV) loss values of Gradient Descent (GD), Adam and PitStop (PS) on the Burgers' equation after 100 iterations.

| Run | GD-PI | GD-SV | Adam-PI | Adam-SV | PS-PI | PS-SV |
|-----|-------|-------|---------|---------|-------|-------|
| 1 | 4.5e-04 | 3.0e-01 | 2.0e-04 | 1.7e-01 | 6.4e-04 | 3.6e-02 |
| 2 | 8.7e-04 | 6.6e-01 | 3.0e-04 | 2.6e-01 | 7.1e-04 | 4.0e-02 |
| 3 | 1.5e-03 | 1.1e+00 | 5.3e-04 | 4.6e-01 | 7.4e-04 | 4.1e-02 |
| 4 | 1.3e-03 | 8.6e-01 | 1.0e-03 | 8.8e-01 | 7.4e-04 | 4.1e-02 |
| 5 | 4.5e-04 | 2.2e-01 | 2.6e-04 | 9.9e-02 | 4.4e-04 | 1.0e-02 |
| 6 | 8.5e-04 | 5.3e-01 | 3.7e-04 | 1.9e-01 | 4.8e-04 | 1.2e-02 |
| 7 | 1.5e-03 | 9.0e-01 | 5.7e-04 | 3.7e-01 | 5.1e-04 | 1.3e-02 |
| 8 | 1.3e-03 | 7.1e-01 | 1.1e-03 | 7.5e-01 | 5.2e-04 | 1.4e-02 |
| 9 | 3.0e-04 | 2.6e-01 | 8.2e-05 | 1.0e-01 | 1.2e-04 | 2.0e-03 |
| 10 | 6.9e-04 | 5.8e-01 | 1.1e-04 | 1.3e-01 | 1.5e-04 | 2.2e-03 |
| 11 | 1.3e-03 | 9.6e-01 | 3.2e-04 | 3.4e-01 | 1.7e-04 | 2.6e-03 |
| 12 | 1.1e-03 | 7.6e-01 | 8.2e-04 | 7.6e-01 | 1.7e-04 | 2.5e-03 |
| 13 | 2.2e-04 | 1.4e-01 | 6.5e-05 | 2.9e-02 | 1.8e-04 | 3.5e-03 |
| 14 | 5.7e-04 | 4.1e-01 | 1.2e-04 | 8.6e-02 | 2.1e-04 | 5.0e-03 |
| 15 | 1.1e-03 | 7.4e-01 | 2.9e-04 | 2.6e-01 | 2.3e-04 | 5.4e-03 |
| 16 | 9.7e-04 | 5.7e-01 | 7.3e-04 | 5.9e-01 | 2.3e-04 | 5.8e-03 |
| 17 | 3.2e-04 | 2.7e-01 | 8.6e-05 | 1.3e-01 | 1.6e-04 | 3.2e-03 |
| 18 | 7.1e-04 | 6.0e-01 | 1.0e-04 | 1.5e-01 | 1.8e-04 | 3.4e-03 |
| 19 | 1.3e-03 | 9.8e-01 | 3.4e-04 | 3.6e-01 | 2.0e-04 | 3.2e-03 |
| 20 | 1.1e-03 | 7.8e-01 | 8.4e-04 | 7.8e-01 | 2.0e-04 | 2.5e-03 |
| Mean | 8.9e-04 | 6.1e-01 | 4.1e-04 | 3.5e-01 | 3.5e-04 | 1.2e-02 |
| Std | 4.0e-04 | 2.7e-01 | 3.2e-04 | 2.6e-01 | 2.2e-04 | 1.4e-02 |

its final configuration in less iteration steps. PitStop does not minimize the physics-informed loss as much as the other methods but its solution is more optimal in terms of the supervised loss.

Table 2: Physics-informed (PI) and (SV) loss values of Gradient Descent (GD), Adam and PitStop (PS) on the Burgers' equation after 1000 iterations.

| Run | GD-PI | GD-SV | Adam-PI | Adam-SV | PS-PI | PS-SV |
|-----|-------|-------|---------|---------|-------|-------|
| 1 | 2.2e-04 | 1.9e-01 | 1.8e-04 | 9.8e-02 | 4.0e-04 | 4.5e-03 |
| 2 | 3.3e-04 | 3.0e-01 | 1.8e-04 | 9.8e-02 | 4.0e-04 | 4.5e-03 |
| 3 | 4.2e-04 | 3.7e-01 | 1.9e-04 | 1.3e-01 | 4.0e-04 | 4.5e-03 |
| 4 | 3.7e-04 | 3.3e-01 | 2.2e-04 | 2.0e-01 | 4.0e-04 | 4.5e-03 |
| 5 | 2.9e-04 | 1.4e-01 | 2.1e-04 | 2.1e-02 | 3.9e-04 | 5.4e-03 |
| 6 | 3.9e-04 | 2.3e-01 | 2.1e-04 | 2.1e-02 | 3.9e-04 | 5.4e-03 |
| 7 | 4.5e-04 | 2.8e-01 | 2.1e-04 | 2.4e-02 | 3.9e-04 | 5.4e-03 |
| 8 | 4.1e-04 | 2.5e-01 | 2.2e-04 | 5.9e-02 | 3.9e-04 | 5.4e-03 |
| 9 | 1.2e-04 | 1.4e-01 | 6.6e-05 | 6.7e-02 | 8.2e-05 | 1.8e-03 |
| 10 | 1.6e-04 | 2.0e-01 | 6.5e-05 | 6.6e-02 | 8.2e-05 | 1.8e-03 |
| 11 | 2.0e-04 | 2.4e-01 | 6.5e-05 | 7.0e-02 | 8.2e-05 | 1.8e-03 |
| 12 | 1.7e-04 | 2.1e-01 | 6.5e-05 | 7.1e-02 | 8.2e-05 | 1.8e-03 |
| 13 | 9.9e-05 | 7.1e-02 | 4.6e-05 | 1.6e-02 | 1.5e-04 | 2.9e-03 |
| 14 | 1.7e-04 | 1.4e-01 | 4.6e-05 | 1.6e-02 | 1.5e-04 | 2.9e-03 |
| 15 | 2.2e-04 | 1.8e-01 | 4.5e-05 | 1.4e-02 | 1.5e-04 | 2.9e-03 |
| 16 | 2.0e-04 | 1.6e-01 | 6.1e-05 | 1.8e-02 | 1.5e-04 | 2.9e-03 |
| 17 | 1.1e-04 | 1.5e-01 | 6.4e-05 | 7.1e-02 | 1.1e-04 | 1.9e-03 |
| 18 | 1.8e-04 | 2.2e-01 | 6.6e-05 | 7.1e-02 | 1.1e-04 | 1.9e-03 |
| 19 | 2.3e-04 | 2.6e-01 | 6.3e-05 | 7.1e-02 | 1.1e-04 | 1.9e-03 |
| 20 | 1.9e-04 | 2.3e-01 | 7.1e-05 | 9.4e-02 | 1.1e-04 | 1.9e-03 |
| Mean | 2.5e-04 | 2.1e-01 | 1.2e-04 | 6.5e-02 | 2.3e-04 | 3.3e-03 |
| Std | 1.1e-04 | 7.1e-02 | 7.1e-05 | 4.5e-02 | 1.4e-04 | 1.4e-03 |

Table 3: Physics-informed (PI) and (SV) loss values of Gradient Descent (GD), Adam and PitStop (PS) on the Burgers' equation after 10000 iterations.

| Run | GD-PI | GD-SV | Adam-PI | Adam-SV | PS-PI | PS-SV |
|-----|-------|-------|---------|---------|-------|-------|
| 1 | 1.9e-04 | 1.3e-01 | 1.8e-04 | 9.8e-02 | 4.0e-04 | 4.5e-03 |
| 2 | 2.0e-04 | 1.7e-01 | 1.8e-04 | 9.8e-02 | 4.0e-04 | 4.5e-03 |
| 3 | 2.1e-04 | 1.8e-01 | 1.8e-04 | 9.8e-02 | 4.0e-04 | 4.5e-03 |
| 4 | 2.0e-04 | 1.7e-01 | 1.8e-04 | 9.8e-02 | 4.0e-04 | 4.5e-03 |
| 5 | 2.1e-04 | 3.5e-02 | 2.1e-04 | 2.1e-02 | 3.9e-04 | 5.4e-03 |
| 6 | 2.1e-04 | 4.1e-02 | 2.1e-04 | 2.1e-02 | 3.9e-04 | 5.4e-03 |
| 7 | 2.2e-04 | 4.4e-02 | 2.1e-04 | 2.1e-02 | 3.9e-04 | 5.4e-03 |
| 8 | 2.1e-04 | 4.1e-02 | 2.1e-04 | 2.1e-02 | 3.9e-04 | 5.4e-03 |
| 9 | 6.5e-05 | 7.1e-02 | 6.5e-05 | 6.6e-02 | 8.2e-05 | 1.8e-03 |
| 10 | 6.5e-05 | 7.2e-02 | 6.6e-05 | 6.6e-02 | 8.2e-05 | 1.8e-03 |
| 11 | 6.5e-05 | 7.2e-02 | 6.5e-05 | 6.6e-02 | 8.2e-05 | 1.8e-03 |
| 12 | 6.5e-05 | 7.2e-02 | 6.5e-05 | 6.6e-02 | 8.2e-05 | 1.8e-03 |
| 13 | 5.7e-05 | 1.1e-02 | 4.5e-05 | 1.6e-02 | 1.5e-04 | 2.9e-03 |
| 14 | 6.1e-05 | 1.7e-02 | 4.6e-05 | 1.6e-02 | 1.5e-04 | 2.9e-03 |
| 15 | 6.1e-05 | 1.8e-02 | 4.7e-05 | 1.6e-02 | 1.5e-04 | 2.9e-03 |
| 16 | 6.1e-05 | 1.7e-02 | 4.5e-05 | 1.6e-02 | 1.5e-04 | 2.9e-03 |
| 17 | 6.9e-05 | 8.9e-02 | 6.3e-05 | 7.1e-02 | 1.1e-04 | 1.9e-03 |
| 18 | 7.2e-05 | 9.7e-02 | 6.4e-05 | 7.1e-02 | 1.1e-04 | 1.9e-03 |
| 19 | 7.2e-05 | 9.8e-02 | 6.3e-05 | 7.1e-02 | 1.1e-04 | 1.9e-03 |
| 20 | 7.2e-05 | 9.7e-02 | 6.4e-05 | 7.1e-02 | 1.1e-04 | 1.9e-03 |
| Mean | 1.2e-04 | 7.7e-02 | 1.1e-04 | 5.4e-02 | 2.3e-04 | 3.3e-03 |
| Std | 7.0e-05 | 5.1e-02 | 6.8e-05 | 3.1e-02 | 1.4e-04 | 1.4e-03 |

Table 4: Physics-informed (PI) and (SV) loss values of Gradient Descent (GD), Adam and PitStop (PS) on the Navier-Stokes equations after 100 iterations.

| Run | GD-PI | GD-SV | Adam-PI | Adam-SV | PS-PI | PS-SV |
|---|---|---|---|---|---|---|
| 1 | 6.5e-03 | 3.1e-01 | 1.3e-03 | 1.3e-01 | 2.1e-03 | 2.5e-03 |
| 2 | 7.4e-03 | 3.1e-01 | 1.0e-03 | 1.2e-01 | 1.9e-03 | 2.5e-03 |
| 3 | 4.8e-02 | 2.6e+00 | 2.5e-02 | 1.6e+00 | 3.4e-03 | 3.5e-03 |
| 4 | 5.0e-02 | 2.6e+00 | 2.5e-02 | 1.4e+00 | 2.8e-03 | 3.2e-03 |
| 5 | 4.9e-01 | 2.8e+01 | 2.8e+00 | 3.8e+01 | 2.1e-02 | 1.6e-02 |
| 6 | 4.9e-01 | 2.8e+01 | 2.5e+00 | 3.5e+01 | 1.1e-02 | 1.2e-02 |
| 7 | 7.4e-03 | 3.1e-01 | 1.0e-03 | 1.2e-01 | 2.0e-03 | 2.6e-03 |
| 8 | 6.5e-03 | 3.1e-01 | 1.3e-03 | 1.3e-01 | 2.1e-03 | 2.6e-03 |
| 9 | 5.1e-02 | 2.6e+00 | 2.6e-02 | 1.5e+00 | 3.2e-03 | 3.4e-03 |
| 10 | 4.8e-02 | 2.6e+00 | 2.4e-02 | 1.5e+00 | 3.2e-03 | 3.5e-03 |
| 11 | 5.0e-01 | 2.8e+01 | 2.7e+00 | 3.8e+01 | 2.1e-02 | 1.6e-02 |
| 12 | 4.8e-01 | 2.8e+01 | 2.7e+00 | 3.6e+01 | 1.2e-02 | 1.3e-02 |
| Mean | 1.8e-01 | 1.0e+01 | 9.0e-01 | 1.3e+01 | 7.2e-03 | 6.7e-03 |
| Std | 2.2e-01 | 1.3e+01 | 1.3e+00 | 1.7e+01 | 7.0e-03 | 5.4e-03 |

Table 5: Physics-informed (PI) and (SV) loss values of Gradient Descent (GD), Adam and PitStop (PS) on the Navier-Stokes equations after 1000 iterations.

| Run | GD-PI | GD-SV | Adam-PI | Adam-SV | PS-PI | PS-SV |
|---|---|---|---|---|---|---|
| 1 | 9.1e-04 | 1.6e-01 | 3.7e-04 | 1.1e-02 | 8.3e-04 | 8.1e-04 |
| 2 | 9.1e-04 | 1.5e-01 | 3.2e-04 | 1.0e-02 | 8.3e-04 | 8.1e-04 |
| 3 | 2.2e-03 | 1.3e+00 | 6.7e-04 | 1.2e-01 | 8.3e-04 | 8.1e-04 |
| 4 | 2.2e-03 | 1.3e+00 | 5.2e-04 | 9.8e-02 | 8.3e-04 | 8.2e-04 |
| 5 | 1.7e-02 | 1.4e+01 | 1.4e-02 | 7.0e+00 | 8.4e-04 | 8.2e-04 |
| 6 | 1.7e-02 | 1.4e+01 | 9.0e-03 | 3.7e+00 | 8.5e-04 | 8.6e-04 |
| 7 | 9.1e-04 | 1.5e-01 | 5.2e-04 | 1.0e-02 | 8.3e-04 | 8.3e-04 |
| 8 | 9.1e-04 | 1.6e-01 | 3.6e-04 | 1.1e-02 | 8.3e-04 | 8.3e-04 |
| 9 | 2.2e-03 | 1.3e+00 | 5.2e-04 | 1.2e-01 | 8.3e-04 | 8.4e-04 |
| 10 | 2.2e-03 | 1.3e+00 | 5.9e-04 | 1.0e-01 | 8.3e-04 | 8.3e-04 |
| 11 | 1.7e-02 | 1.4e+01 | 1.4e-02 | 7.0e+00 | 8.5e-04 | 8.8e-04 |
| 12 | 1.7e-02 | 1.4e+01 | 8.7e-03 | 3.7e+00 | 8.5e-04 | 8.6e-04 |
| Mean | 6.6e-03 | 5.2e+00 | 4.1e-03 | 1.8e+00 | 8.4e-04 | 8.3e-04 |
| Std | 7.2e-03 | 6.3e+00 | 5.3e-03 | 2.7e+00 | 9.8e-06 | 2.2e-05 |

Table 6: Physics-informed (PI) and (SV) loss values of Gradient Descent (GD), Adam and PitStop (PS) on the Navier-Stokes equations after 10000 iterations.

| Run | GD-PI | GD-SV | Adam-PI | Adam-SV | PS-PI | PS-SV |
|---|---|---|---|---|---|---|
| 1 | 4.1e-04 | 4.6e-02 | 2.5e-04 | 8.9e-03 | 6.4e-04 | 5.5e-04 |
| 2 | 4.1e-04 | 4.4e-02 | 2.6e-04 | 8.9e-03 | 6.4e-04 | 5.5e-04 |
| 3 | 5.9e-04 | 3.0e-01 | 4.1e-04 | 8.9e-03 | 6.4e-04 | 5.5e-04 |
| 4 | 5.9e-04 | 2.9e-01 | 7.4e-04 | 9.0e-03 | 6.4e-04 | 5.5e-04 |
| 5 | 2.5e-03 | 3.1e+00 | 3.4e-04 | 8.8e-03 | 6.4e-04 | 5.5e-04 |
| 6 | 2.6e-03 | 3.2e+00 | 2.7e-04 | 8.8e-03 | 6.4e-04 | 5.5e-04 |
| 7 | 4.1e-04 | 4.3e-02 | 3.3e-04 | 8.5e-03 | 6.4e-04 | 5.6e-04 |
| 8 | 4.1e-04 | 4.6e-02 | 2.5e-04 | 8.5e-03 | 6.4e-04 | 5.5e-04 |
| 9 | 5.8e-04 | 2.9e-01 | 2.8e-04 | 8.5e-03 | 6.4e-04 | 5.6e-04 |
| 10 | 5.9e-04 | 3.0e-01 | 2.5e-04 | 8.5e-03 | 6.4e-04 | 5.5e-04 |
| 11 | 2.5e-03 | 3.0e+00 | 5.0e-04 | 8.4e-03 | 6.4e-04 | 5.6e-04 |
| 12 | 2.6e-03 | 3.2e+00 | 4.4e-04 | 8.5e-03 | 6.4e-04 | 5.5e-04 |
| Mean | 1.2e-03 | 1.2e+00 | 3.6e-04 | 8.7e-03 | 6.4e-04 | 5.5e-04 |
| Std | 9.8e-04 | 1.4e+00 | 1.4e-04 | 2.1e-04 | 1.3e-06 | 3.6e-06 |

