# OpenReview forum: "PitStop: Physics-Informed Training with Gradient Stopping"
_ICLR.cc/2026/Conference — Submitted to ICLR 2026_

### Official Review · Reviewer_GPk5 · 2025-10-21

**Soundness:** 3
**Presentation:** 3
**Contribution:** 2
**Rating:** 4
**Confidence:** 4

**Summary:**

This paper introduces PitStop to optimize physics-informed loss that modifies gradient backpropagation by stopping certain gradient flows, thereby deviating from the standard chain rule. The authors argue that classical optimization methods are ill-suited for physics-informed tasks because the residual-based loss functions are ill-conditioned and their approximate minima often fail to align with the supervised objective. PitStop is theoretically grounded using linear fixed-point theory and is shown to ensure convergence under mild conditions. The method requires no extra computational cost compared to standard gradients and exhibits improved convergence rates and stability. Extensive experiments on linear and nonlinear systems, including Burgers’ and Navier–Stokes equations, demonstrate faster convergence and better supervised accuracy compared to standard optimizers.

**Strengths:**

1.The paper provides a comprehensive framework that connects optimization theory and physics-informed learning, especially through non-symmetric fixed-point operators and gradient stopping.

2.Theorems on convergence, rate, and fixed points are well-structured and backed by detailed proofs.

3.Reinterpreting backpropagation itself, rather than tuning weights or loss terms, is an innovative contribution to the field.

4.The method maintains computational efficiency, making it suitable for large-scale physics-informed neural network (PINN) problems.

**Weaknesses:**

1.The analysis is confined primarily to linearized systems and time/space-discretized settings, raising uncertainty about applicability to fully nonlinear PDEs or continuous-time PINNs.

2.The method requires explicit temporal discretization, potentially limiting use in frameworks that rely on automatic differentiation.

3. While the proofs seems rigorous, the implementation details and intuition behind gradient stopping, as well as some symbols in the proofs could be clarified further for non-theoretical audiences.

4. The effect of hyperparameters (e.g., $\alpha$ , learning rate $\eta$) on convergence robustness could be elaborated.

**Questions:**

1.Can the convergence guarantees extend to nonlinear PDE operators or models with data-driven mixed losses?

2.How sensitive is the method to the choice of α (initial condition weight) and step size η in practice?

3.The physics-informed loss (6) is presented in a discretized form, i.e. in the settings the temporal and spatial domains are setting to equidistant points. This is too restrict. As we know, one of PINN’s advantage is its meshless property, but (6) is the form of finite difference scheme loss, reduced the advantage of the continuous physics-informed loss. In this setting we can solve (6) by solving linear algebraic equations like in the finite difference method, so why we need another optimization method to solve (6)? How to extend the method to nonregular domain? Can the method extend to continuous time/space physics-informed loss?

---

> ### Author Response · Authors · 2025-11-21
>
> Dear Reviewer,
>
> Thank you for your review. We hope these answers to your questions will address your concerns:
>
> Q1) “Can the convergence guarantees extend to nonlinear PDE operators or models with data-driven mixed losses? ”
>
> The purpose of our experiments with nonlinear PDEs was to gain a first indication of whether it might be possible to derive a convergence guarantee for such systems. Based on the observation that PitStop converged successfully without extensive tuning, we suspect that this may indeed be feasible, potentially under additional assumptions.
>
> The generalization of our convergence guarantee to losses that combine data-driven and physics-informed terms is straightforward: the iteration operators corresponding to each individual loss are positive definite, and therefore their sum is also positive definite.
>
>
>
> Q2) “How sensitive is the method to the choice of $\alpha$ (initial condition weight) and step size $\eta$ in practice?”
>
>
>
>
> In practice, an $\alpha$ value of $1.0$, which is the natural choice, did not automatically lead to divergence, as observed in our experiments on the nonlinear systems. We reported the condition that
> $\alpha$ must satisfy in order to present mathematically precise theorems.
>
> Regarding the step size $\eta$, the dependence is stable in the sense that there is a large range of possible values for which the method converges slower but is still convergent. A technical but interesting detail from the theory is that the convergence rate around the optimal step size can have a parabolic shape for PitStop, due to its nonsymmetric iteration operator. For Gradient Descent, the convergence rate as a function of step size has a piecewise-linear shape. To illustrate this, we included a plot in the appendix of our revised draft showing the measured convergence rate as a function of step size on the linear oscillator system, which demonstrates both this phenomenon and the general dependence of convergence rate on step size.

---

> > ### Author Response · Authors · 2025-11-21
> >
> > Q3) “The physics-informed loss (6) is presented in a discretized form, i.e. in the settings the temporal and spatial domains are setting to equidistant points. This is too restrict. As we know, one of PINN’s advantage is its meshless property, but (6) is the form of finite difference scheme loss, reduced the advantage of the continuous physics-informed loss. In this setting we can solve (6) by solving linear algebraic equations like in the finite difference method, so why we need another optimization method to solve (6)? How to extend the method to nonregular domain? Can the method extend to continuous time/space physics-informed loss?”
> >
> > It is correct that our work operates under two main assumptions:
> >
> > First, discretization of the temporal derivatives. This corresponds to the discrete-time PINN in the original PINN paper. Our method requires this assumption because the residual terms must involve two network predictions at different points in time for the method to be well-defined and for the convergence proof to hold. At present, we do not know whether this idea can be generalized to continuous-time PINNs, and we acknowledge this limitation in the conclusion of our paper. It is worth noting, however, that this is equally a limitation for continuous-time PINNs if it is indeed not possible and no other efficient optimization approaches exist for them.
> >
> > Second, fixed grid points. This assumption is primarily made to simplify the analysis. Without it, the grid points would be resampled at each iteration, causing the loss to change continuously. Instead of the smooth loss curves observed for the harmonic oscillator, we would see noisier and more fluctuating curves, making it harder to illustrate our theoretical statements and draw conclusions. On a technical level, we would also need to work with integrals, functionals, and functional analysis instead of sums, vectors, and linear algebra. This adds significant mathematical complexity without substantially increasing insight. We believe that the reduced case is better suited to present our ideas even to readers who are less theory-focused.
> >
> >
> > Extensions of PitStop to nonregular grids can be achieved by following the procedure outlined in our paper: define a candidate method, assume linear function approximation, determine the specific form of the fixed-point operator, and check for positive definiteness, which essentially corresponds to Theorems 1 and 2. The remaining theorems are more general and follow directly from the use of gradient stopping and the structure of the physics-informed setup.
> >
> > Finally, the question "why do we need another optimization method for linear systems?" This brings us back to the core of our paper. All existing methods approach the minimum of the physics-informed loss. In large-scale scenarios, where it is impossible to represent every grid point exactly and function approximation is necessary, this minimum can be far from the supervised minimum. We believe the features of our method---computational efficiency, theoretical foundation, and experimental validation---are sufficiently interesting to motivate further research in this direction.

---

> > > ### Comment · Reviewer_GPk5 · 2025-11-24
> > >
> > > Thanks for the response. My question in Q3 is not well-addressed. Most PINN's work follow the continuous-time framework, the collocation points are randomly sampled in space and time, they may even different in different iterations. The extension to the continuous-time framework is unclear at present.
> > >
> > > On the other hand, if we can set the temporal and spatial domains to equidistant points, then it can be well solved by traditional numerical methods. At present, no evidence shows the advantage over numerical methods in this case.

---

### Official Review · Reviewer_8BHV · 2025-10-27

**Soundness:** 2
**Presentation:** 1
**Contribution:** 2
**Rating:** 2
**Confidence:** 4

**Summary:**

This paper provides an interesting study on the effect of gradient stopping in the optimization of time-dependent PINN loss. The authors propose to stop gradient propagation during autoregressive residual calculation. The experimental results show the benefit of this approach.

**Strengths:**

This work performs systematic study of the effect of gradient stopping in training autoregressive PDE solvers, a popular trick that hasn't been properly investigated before. The authors analyze the effect of gradient stopping in linearized fix point setting, which is simple but illustrative perspective. The experimental results shows the effectiveness of this approach comparing to default optimizers.

**Weaknesses:**

I have a few main concerns:

1. The problem setting of PINN is non-standard. The authors say "We restrict ourselves to a setup where the time derivative is discretized directly through a numerical time-stepping scheme, rather than being computed via automatic differentiation". For most PINNs, there's no time-stepping and time derivatives are calculated very efficiently using AD. Even for PINOs, time derivatives can be evaluated using higher-order scheme.

2. The comparison to standard PINNs is missing. It is likely that the mismatch of GD is due to the way time derivatives are calculated.

**Questions:**

1. Is this PINN setting general enough?

2. The experiments lack basic setting. What is the input/output of the network? If it's a standard MLP with time input, how is the time-stepping done? If not, how many roll out steps are done for both methods?

---

> ### Author Response · Authors · 2025-11-21
>
> Dear Reviewer,
>
> Thank you for your review. We hope these answers to your questions will address your concerns:
>
> Q1)”Is this PINN setting general enough?”
>
> As you mentioned, there is a different class of PINNs, called continuous-time PINNs in the original PINN paper, which use automatic differentiation. We would not necessarily call their computation of the time derivatives more efficient as it comes at the cost of a larger computation graph when the loss computation already contains backpropagation passes for the time derivatives.
>
> While we are interested if our method can be generalized to continuous-time PINNs, our theoretical analysis applies so far only to discrete-time PINNs. Therefore, it makes sense to restrict the setup accordingly; it is the most general setup for which our analysis applies. Because the convergence mechanism is fundamentally different from gradient methods without additional computational cost, we hope this work can be a stepping stone for similar methods for continuous-time PINNs.
>
>
> Q2)"The experiments lack basic setting. What is the input/output of the network? If it's a standard MLP with time input, how is the time-stepping done? If not, how many roll out steps are done for both methods?"
>
> We added further explanations at the beginning of the experiments section, summarizing the main points from our “Problem Setup” section to help clarify the experimental setups. In our setups, the function approximator directly represents the physical fields $y(x,t)$, with inputs given by points in space and time and outputs corresponding to the field value at that position. While rollouts are formally included in this formulation as a special case, they were neither the focus of our analysis nor used in the experiments.
>
> We discuss the details of the function approximator in the appendix, as they are not essential for understanding the main insights of the paper, which focus on the optimization procedure rather than architectural choices. For the same reason, we did not include a comparison with continuous-time PINNs, since they involve a different loss function. To properly assess the effects of different optimizers, it is important to start from the same loss function.

---

### Official Review · Reviewer_mAoF · 2025-10-27

**Soundness:** 3
**Presentation:** 3
**Contribution:** 3
**Rating:** 4
**Confidence:** 4

**Summary:**

This paper proposes new optimization solutions for the key challenges of PINN optimization. The problem solved is very meaningful, and the method is also novel, but the insufficient evaluation of the results has affected the performance of the method. If the author solves my concerns and demonstrates that the proposed method has significant advantages, I will increase the rating.

**Strengths:**

1. A new physics-informed loss optimization method has been derived through theoretical derivation, which has theoretical basis.
2. Verified on 4 PDEs.
3. The problem being solved is urgent and important.

**Weaknesses:**

1. In Line 97, why can such an assumption be made? This approach yields different results from automatic differentiation, and different discretization methods can also lead to different outcomes.
2. A typo is in Line 131.
3. The experimental section only compares one curve. If experiments can be conducted under different boundary conditions and initial conditions, and statistical analysis can be added, it will improve the credibility of the paper.
4. PINN will use second-order optimizers such as LBFGS after Adam, but this baseline is not mentioned in the paper.
5. The paper lacks an introduction to the baseline. Is the calculation of PDE loss done using discrete methods or automatic differentiation in time derivatives?
6. Can you draw an error graph for Figure 4, as it appears that gradient descent is more similar to solution.
7. The paragraph starting from line 474 appears to have no relation to the previous part, and there is no relevant evidence before it. This paragraph makes the paper unscientific.

**Questions:**

See the weaknesses.

---

> ### Author Response · Authors · 2025-11-21
>
> Dear Reviewer,
>
> Thank you for your feedback. We hope our comments will address the mentioned weaknesses:
>
> 1) “In Line 97, why can such an assumption be made? This approach yields different results from automatic differentiation, and different discretization methods can also lead to different outcomes.”
>
> Correct, these approaches are all different formulations of a loss function used to approximate a solution to a PDE problem. It is important to note that even a formulation based on autodiff will only yield an approximate solution (unless the exact solution is already known and the function approximator is specifically constructed to represent it, which is a practically irrelevantcase). The assumption is primarily made to allow us to present a novel theoretical perspective on training with physics-informed loss functions.
>
> 2) “A typo is in Line 131.”
>
> Fixed.
>
> 3) “The experimental section only compares one curve. If experiments can be conducted under different boundary conditions and initial conditions, and statistical analysis can be added, it will improve the credibility of the paper.”
>
> Thank you for the suggestion. One reason for displaying individual curves was to illustrate the relationships between the supervised and physics-informed losses. We felt that non-averaged learning curves might be more appropriate in this context.
>
> Nevertheless, we conducted additional experiments for the nonlinear PDEs, as these scenarios extend beyond the scope of our theoretical analysis. For example, in the case of Burgers’ equation, we repeated our experiments with five different boundary conditions and four different initializations. The outcomes of all runs are reported in the paper. In summary, the mean loss values after 10,000 iterations are as follows:
>
> Gradient Descent, physics-informed loss: $1.2\cdot 10^{-4}$
>
> Gradient Descent, supervised loss: $7.7\cdot 10^{-2}$
>
> Adam, physics-informed loss: $1.1\cdot 10^{-4}$
>
> Adam, supervised loss: $5.4\cdot 10^{-2}$
>
> PitStop, physics-informed loss: $2.3\cdot 10^{-4}$
>
> PitStop, supervised loss: $3.3\cdot 10^{-3}$
>
>
> These results reflect the main insights of our paper. Classical methods, such as Gradient Descent and Adam, tend to approach the minimum of the physics-informed loss, which is suboptimal with respect to the supervised loss. In contrast, PitStop does not minimize the physics-informed loss as aggressively, but its solutions achieve better supervised loss values. For the Navier–Stokes equations, we conducted similar experiments and observed the same pattern. Complete data for all experiments are provided in the appendix of our revised paper.
>
>
> 4) “PINN will use second-order optimizers such as LBFGS after Adam, but this baseline is not mentioned in the paper.”
>
> L-BFGS is another method that can accelerate convergence to the (local) least-squares solution of a loss function. Since Adam was already able to reach this solution, we saw no need to include L-BFGS.
>
> It is worth noting that, like Newton’s method, L-BFGS is computationally more expensive, requiring multiple function and gradient evaluations per iteration. This was precisely why we replaced Gauss–Newton with Adam when moving from the oscillator example to Burgers’ equation and Navier–Stokes. In contrast to both of them, our method, like Adam and Gradient Descent, requires only one forward and one backward pass per iteration.
>
> 5) “The paper lacks an introduction to the baseline. Is the calculation of PDE loss done using discrete methods or automatic differentiation in time derivatives?”
>
> We study exclusively the effects of the optimization procedure. For that reason, the loss formulation for each method (Gradient Descent, Gauss-Newton, Adam, PitStop) is the same on each physical system. In all cases, time derivatives were discretized. We added additional explanations of the experimental setup at the beginning of the experiments.
>
> 6) “Can you draw an error graph for Figure 4, as it appears that gradient descent is more similar to solution.”
>
> There was an error in the plot, which displayed the solution again under “Gradient Descent.” We have corrected this, and the updated plot now shows the actual results from Gradient Descent, which clearly differ from the solution.
>
>
> 7) “The paragraph starting from line 474 appears to have no relation to the previous part, and there is no relevant evidence before it. This paragraph makes the paper unscientific.”
>
> Thank you for pointing this out. We removed the speculative claims.

---

> > ### Comment · Reviewer_mAoF · 2025-11-26
> >
> > 1. What is the author's answer to the first question? What I would like to know is the reason. If the author thinks it is correct, should it be modified in the paper?
> >
> > 2. I cannot find the author's modifications in the paper, such as the third answer. If any modifications have been made, please indicate them in other colors.
> >
> > 3. The cost of L-BFGS is indeed high, but it is a common PINN optimizer that the author should at least compare with and prove to be better than L-BFGS.
> >
> > 4. Even if all methods use the same loss calculation method, the current results can only prove that the proposed method is good in a single form and cannot explain more.
> >
> > 5. What is the reason for this error in the sixth question?
> >
> > I did not see any modifications in the paper, so I will maintain the rating.

---

### Official Review · Reviewer_WCm7 · 2025-10-31

**Soundness:** 2
**Presentation:** 2
**Contribution:** 3
**Rating:** 4
**Confidence:** 2

**Summary:**

This paper presents an optimization framework named PitStop for optimizing physics-informed objective functions. In the current paradigm, most work optimizes Physics-informes loss, which corresponds to the supervision loss of the temporal derivative of the governing physics model. By optimizing this derivative-based loss, the existing methods aim at attaining the optimal Supervision loss. However, this work points out that the optimal points of these two different loss functions are not always the same, and based on this observation, proposes PitStop as an alternative for the existing classical gradient-based optimization methods like Gradient Descent and Gauss-Newton methods. To analyze the properties of the PitStop, the authors use the lens of linear fixed-point iterations. With this interpretation, the authors claim that while PitStop could converge to the worse fixed point than the Gradient Descent in terms of the physics-informed loss, it eventually converges to the better Supervision loss faster than these methods. The authors provide experimental results on one toy example, harmonic oscillator, Burger's equation, and Navier-Stokes equation, and show that PitStop converges faster to the better solution than the conventional methods.

**Strengths:**

- The motivation is good, and authors provided a thorough theoretical analysis of their approach.
- With simple experiments, the authors effectively show that the current approach to minimize the Physics-informed loss does not always align with the final goal to minimize the Supervision loss. They also sho

**Weaknesses:**

- The overall description was hard to follow. More intuitive explanation about why PitStop works would be appreciated.
- The overall description was hard to follow, making it challenge to reproduce the results
- Even though the authors gave detailed definitions and analysis of their approach, it is unclear how we can implement it. Seeing the equation 7 and 8, I feel like we can reproduce the results by only cutting the gradient flow across different time steps, but I'm not sure if I understand it correctly. It would be helpful if the authors provide an explicit algorithm (or pseudo code).
- The experimental results are not convincing that this approach is an overall better approach than existing methods.  For example,  in   Figure 4, I believe (d) GD gives better result than PitStop.

**Questions:**

See above.

---

> ### Author Response · Authors · 2025-11-21
>
> Dear Reviewer,
>
> Thank you for your feedback. We hope our comments will address the mentioned weaknesses:
>
> 1) “The overall description was hard to follow. More intuitive explanation about why PitStop works would be appreciated.”
>
> PitStop can be understood from a causal perspective. In initial value problems, earlier states influence later ones, not the reverse. Consequently, when a physics-informed residual is nonzero and the later state in that residual term fails to match the dynamics implied by the earlier state in that residual, then the optimization algorithm should modify only the later state. This is what PitStop does through gradient stopping. We added a paragraph after we introduce our method to add this intuitive view.
>
>
> 2) “The overall description was hard to follow, making it challenge to reproduce the results.”
>
> In addition to the theoretical description of our method, we also provide pseudocode outlining the implementation steps. For the experiments, the details of both function approximation and the physical systems are specified in the appendix, including the values of all system parameters, which should in principle enable reproduction. Furthermore, our reproducibility statement notes that we intend to release our code along with the paper. Together, these materials should allow full reproduction of our results.
>
> 3) “Even though the authors gave detailed definitions and analysis of their approach, it is unclear how we can implement it. Seeing the equation 7 and 8, I feel like we can reproduce the results by only cutting the gradient flow across different time steps, but I'm not sure if I understand it correctly. It would be helpful if the authors provide an explicit algorithm (or pseudo code).”
>
> You have understood our method correctly. Please note that the pseudocode describing our method is provided at the top of page 3. We apologize if the definitions and mathematical statements make it dense to read, but without them it would be impossible to present a theoretical analysis.
>
> 4) “The experimental results are not convincing that this approach is an overall better approach than existing methods. For example, in Figure 4, I believe (d) GD gives better result than PitStop.”
>
> There was an error in the plot, which displayed the solution again under “Gradient Descent.” We have corrected this, and the updated plot now shows the actual results from Gradient Descent, which clearly differ from the solution.

---

### Author Response · Authors · 2025-11-21
**Paper Update**

Dear Reviewers,

Thank you all for reviewing our paper. We have uploaded an updated draft. In addition to minor corrections and further clarifications in the main text, we have added a new section to the appendix (Section C, page 20-23). This section provides additional experimental evaluation on the nonlinear systems (Burgers’ equation and the Navier–Stokes equations). These supplementary results further indicate that our method performs favorably in those cases as well.

---

### Meta-Review · Area_Chair_GJUz · 2026-01-06

**Summary:**

Across the four reviews and the ensuing discussion, the overall signal remains negative: while reviewers appreciate the motivation and the attempt to rethink optimization for physics-informed training. Reviewers view the current contribution as not yet compelling enough for acceptance because the core claims are supported mainly in a restricted, discretized PINN setting. So the recommendation is reject.

**Reviewer Concerns:**

The rebuttal helps on presentation, but several central concerns remain: (i) the method’s generality is still unclear—multiple reviewers question how it extends to the more standard continuous-time PINN framework with AD-based derivatives and randomly sampled collocation points; (ii) the experimental validation is still perceived as incomplete, including missing or disputed baselines and limited controls; and (iii) reviewers raise a broader issue: when the domain is set to equidistant grid points, it is not clearly demonstrated why PitStop is preferable to established numerical solvers or what its advantage is in nonregular domains.

**Reviewer Scores:**

Given the maintained skepticism in the discussion and the fact that the score profile is three marginal-below-threshold ratings (4) plus one clear reject (2), I do not expect meaningful upward score movement; overall, the paper remains below the acceptance threshold.

---

### Decision · Program_Chairs · 2026-01-26

Reject